# Policy Design for Two-sided Platforms with Participation Dynamics

**Haruka Kiyohara** [1]   **Fan Yao** [2]   **Sarah Dean** [1]

## Abstract

In two-sided platforms (e.g., video streaming or e-commerce), viewers and providers engage in interactive dynamics: viewers benefit from increases in provider populations, while providers benefit from increases in viewer population. Despite the importance of such "population effects" on long-term platform health, recommendation policies do not generally take the participation dynamics into account. This paper thus studies the dynamics and recommender policy design on two-sided platforms under the population effects for the first time. Our control- and game-theoretic findings warn against the use of the standard "myopic-greedy" policy and shed light on the importance of provider-side considerations (i.e., effectively distributing exposure among provider groups) to improve social welfare via population growth. We also present a simple algorithm to optimize long-term social welfare by taking the population effects into account, and demonstrate its effectiveness in synthetic and real-data experiments. Our experiment code is available at https://github.com/sdean-group/dynamics-two-sided-market.

## 1. Introduction

Two-sided platforms, where some individuals view content or information provided by other individuals, are ubiquitous in real-world decisions, e.g., video streaming, job matching, and online ads (Boutilier et al., 2023). In such applications, viewers and providers may co-evolve and mutually influence each other: providers increase their content production if they receive more attention from viewers (i.e., exposure), and the platform gains more viewers if viewers receive high-quality and favored content (i.e., satisfaction). These effects are mediated by the platform's recommendation algorithm.

Considering such *non-stationarity* and *two-sided dynamics* is crucial, as the viewers and providers are affected by each others' population in self-reinforcing feedback loops.

**Example 1** (Video recommendation). *When a platform has many videos about sports, viewers can expect that top sports videos have high quality (e.g., production and intellect). Meanwhile, if a platform is popular among sports lovers, creators will produce more sports videos to gain more views.*

**Example 2** (Job matching). *When a platform has many applicants from a target category, companies looking to fill a specific role can identify more highly skilled applicants. On the other hand, if a platform has more openings for a specific job type, more applicants from target categories will register for the service.*

The *"population effects"* in the aforementioned examples strongly affect viewer utility and their long-term satisfaction. However, their implications for recommendation policy design have been under-explored. The conventional formulation of recommendation follows (contextual) bandits (Li et al., 2010) and assumes that viewers and providers are static across timesteps. Some recent work studies content provider departures (Mladenov et al., 2020; Huttenlocher et al., 2023) and the (negative) impacts on viewer welfare (Yao et al., 2024b; 2023a). However, the viewer population is modeled as static. In contrast, existing works which consider dynamic viewer populations assume that provider population is fixed (Hashimoto et al., 2018; Dean et al., 2022). Therefore, we cannot tell how we should optimize a policy, particularly in the initial launch of the platform when two-sided dynamics exist. Finally, existing works considering strategic content providers (Hron et al., 2022; Jagadeesan et al., 2022; Yao et al., 2023b; 2024a; Prasad et al., 2023) model the (strategic) evolution of provider features, assuming that the total number of viewers and providers are fixed. These works cannot tell if a platform can "grow the pie" (i.e., viewer and provider populations) to improve long-term welfare.

In response to this gap, **we study the dynamics of *"population effects"* on two-sided platforms**. Specifically, we consider viewer and provider participation dynamics which operate as follows: (1) the population of providers increases as their exposure increases, (2) the population of viewers increases as their satisfaction increases, and (3) the potential

---

[1]Cornell University [2]University of Virginia. Correspondence to: Haruka Kiyohara <hk844@cornell.edu>, Sarah Dean <sdean@cornell.edu>.

*Proceedings of the $42^{nd}$ International Conference on Machine Learning*, Vancouver, Canada. PMLR 267, 2025. Copyright 2025 by the author(s).

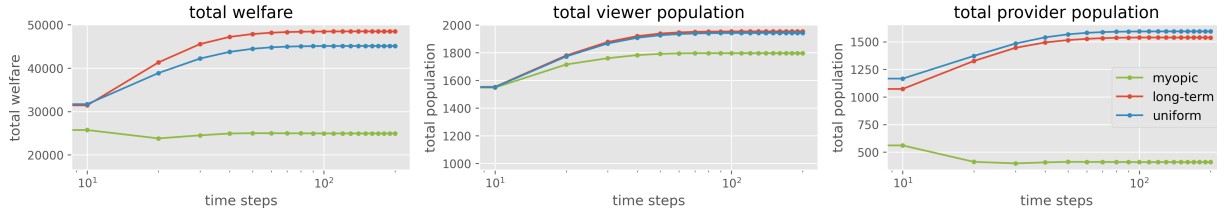

*Figure 1.* **Comparing the myopic-greedy policy, the uniform random policy, and the long-term policy in a synthetic simulation.** As shown, the myopic-greedy policy loses the provider population due to concentrated exposure allocation, resulting in the negative impact on the viewer welfare in the long-run. The "long-term" policy is based on the algorithm proposed in Section 5 (Eq. (10)), and the experiment setting follows Section 6 (with a small initial population).

for viewer satisfaction increases as provider populations increase. We assume that these effects follow an arbitrary monotonically increasing function, and the immediate utility (in the form of exposure or satisfaction) is observed. The key consequence of setting is that the default approach to recommendation can perform much worse in the long term than even a uniform random policy. Figure 1 illustrates this shortcoming of the default approach, a myopic-greedy policy that recommends providers to viewers on the basis of immediate utility.

We examine success and failure cases of the myopic-greedy policy through control- and game-theoretic analyses. Our primary results are the following three points. First, by analyzing the convergence conditions of the dynamics, we argue that concentrated exposure allocation among provider groups can easily cause polarization of viewer and provider populations, potentially resulting in a smaller pie (i.e., populations) and long-term social welfare compared to an exposure-distributing policy. These findings highlight the importance of provider-side awareness such as exposure fairness (Singh & Joachims, 2018) for the long-term success of two-sided platforms under population dynamics. Second, we analyze the linear case to show that the myopic-greedy policy is guaranteed to be optimal only if the population effects (i.e., utility gain by the population growth) are homogeneous across provider groups. Third, we explain the shortcomings of the myopic-greedy policy by decomposing the welfare sub-optimality into two terms: the *"policy regret"* and *"population regret"*. The former comes from the difference between the policy and the myopic-optimal policy at each timestep given the current population, while the latter comes from the difference between the current population and the population under the optimal policy. By definition, the myopic-greedy policy minimizes only the policy regret (i.e., short-term objective). Because the myopic-policy ignores the population regret (i.e., long-term impact on the dynamics), the myopic-greedy policy fails when the scale of the long-term utility gain from the population growth is large.

Finally, we propose a simple algorithm that balances the pol-

icy and population regrets by projecting the long term population that will result from the current viewer satisfaction and provider exposure. The proposed **"look-ahead" policy optimizes the utility at the projected long term population** instead of the immediate population. The synthetic and real-data experiments using the KuaiRec dataset (Gao et al., 2022) demonstrate that the proposed algorithm works better than both myopic-greedy and uniform random policies in multiple configurations by better trading off the long and short term goals accounting for the population growth.

Our contributions are summarized as follows:

- We formulate the "population effects" in two-sided platforms where viewer and provider populations evolve.
- We find that the myopic-greedy policy can fall short when the population effects are heterogeneous.
- We also find that an exposure-guaranteeing policy can be useful for growing populations and minimizing the population regret.
- We propose a simple algorithm that considers the long term population and demonstrate its effectiveness in the synthetic and real-data experiments.

## 2. Viewer-provider two-sided systems

This section models the dynamics of viewer and provider populations on a recommendation platform. Specifically, we consider sub-group dynamics where viewers and providers are categorized into $K$ and $L$ subgroups[1]. Then, we model the populations, recommendation policy, payoffs, and social welfare as follows.

1. (Viewer/provider population) Let $\lambda_k \in \mathbb{R}_{\geq 0}$ be the population of the viewer group $k \in [K]$ and $\lambda_l$ be that of the provider group $l \in [L]$. We also let $\boldsymbol{\lambda} := (\lambda_1, \lambda_2, \cdots, \lambda_K, \lambda_1, \lambda_2, \cdots, \lambda_L)$ be the joint population vector of viewers and providers.

---

[1]We can consider a "subgroup" of size 1. In such cases, the viewer "population" corresponds to the time spent by an individual viewer, while the provider "population" can be the amount of content produced by an individual provider.

2. (Platform's recommendation policy) The platform matches each viewer group $k$ to a provider group $l$ with a recommendation policy denoted by a $K$-by-$L$ matrix $\boldsymbol{\pi}$. Specifically, its $(k, l)$-th element $\pi_{k,l}$ represents the probability of allocating the provider group $l$ to the viewer group $k$. Thus $\sum_{l=1}^{L} \pi_{k,l} = 1, \forall k \in [K]$. For example, the uniform random policy, which assigns equal exposure probability across all provider groups is represented as given by $\boldsymbol{\pi} = \frac{1}{L} \mathbf{1}_{L \times K}$.

3. (Viewer/provider payoffs) After viewer and provider groups are matched by the policy $\boldsymbol{\pi}$, their perceived payoffs can be quantified by the following metrics:

$$\text{Viewer Satisfaction:} \quad s_k = \sum_{l=1}^{L} \pi_{k,l} q_{k,l}, \quad (1)$$

$$\text{Provider Exposure:} \quad e_l = \sum_{k=1}^{K} \pi_{k,l} \lambda_k, \quad (2)$$

where $q_{k,l}$ is the (expected) utility that viewers $k$ receive from the provider group $l$. Eqs. (1) and (2) define viewer satisfaction as determined by the total utility they receive from recommendations, while providers care about the total amount of exposure they receive by recommendation. This model is prevalent is prior works including (Singh & Joachims, 2018; Mladenov et al., 2020).

4. (Social welfare) Finally, we consider the following total viewer welfare as the global metric of the platform:

$$R(\boldsymbol{\pi}; \boldsymbol{\lambda}) := \sum_{k=1}^{K} \lambda_k s_k$$

Note that here we consider the sum of viewer-side satisfaction as the social welfare, a formulation prevalent in related works (Mladenov et al., 2020; Huttenlocher et al., 2023). The sum of content-side exposure simplifies to the total size of the viewer population.

### 2.1. Interaction dynamics and "population effects"

We have so far seen a typical formulation in two-sided platforms. However, a key limitation of such formulation is to ignore potential non-stationarity in the viewer and provider populations, which is common in many real-world two-sided systems (Boutilier et al., 2023; Deffayet et al., 2024).

First, consider the impact of provider population growth on the utility experience by viewers, which we refer to as *"population effects"*. An increase in provider population naturally leads to more high-quality content. For example, consider a two-stage recommendation policy, where our higher-level policy $\boldsymbol{\pi}$ decides the matching between viewer and provider groups, and a second-stage policy selects individual providers among the selected group. Any reasonable second stage policy should be able to select a better provider from a larger provider pool (Su et al., 2023; Evnine et al., 2024). To model such "population effects", we introduce the following utility decomposition:

$$q_{k,l} = b_{k,l} + f_{k,l}(\lambda_l) \quad (3)$$

where $b_{k,l}$ is the *base* utility, which indicates the matching between the preference of viewer and provider groups (e.g., this viewer group likes sports articles). In contrast, $f_{k,l}(\cdot)$ represents the quality of the provider which improves as the provider population increases. $f_{k,l}$ might be heterogeneous among different viewer and provider groups because quality might be multi-dimensional (e.g., visuals, intellects, novelty), viewers may have different preferences, and providers may have different abilities. We take $f_{k,l}$ to be a monotonically increasing function.

Next, consider the impact of viewer and provider payoffs on the population. The number of viewers that a platform can maintain is related to the level of satisfaction; similarly, the number of providers is related to the exposure. We assume that viewer and provider subgroups have some *"reference"* population $\bar{\lambda}_k(s_k)$ and $\bar{\lambda}_l(e_l)$ given the level of viewer satisfaction $s_k$ and provider exposure $e_l$. We assume that $\bar{\lambda}$ is a monotonically increasing function, so higher viewer satisfaction and provider exposure result in increased populations. Based on this, we model the viewer and provider population dynamics as follows:

$$\text{Viewer:} \quad \lambda_{t+1,k} = (1 - \eta_k)\lambda_{t,k} + \eta_k \bar{\lambda}_k(s_{t,k}), \quad (4)$$

$$\text{Content:} \quad \lambda_{t+1,l} = (1 - \eta_l)\lambda_{t,l} + \eta_l \bar{\lambda}_l(e_{t,l}), \quad (5)$$

where $\eta \in [0, 1]$ are the *reactiveness* hyperparams, determining how fast the population changes. Note that similar models are widely adopted in performative predictions (Perdomo et al., 2020; Brown et al., 2022). We thus have that the viewer satisfaction $s_k$ depends on the provider population via "population effects" $f_{k,l}$, while the provider exposure directly depends on the viewer population. The two-sided platform has complex dynamics between viewers and providers. Our goal will be to consider long-term objectives under such co-evolving and two-sided dynamics.

### 2.2. Game-theoretic interpretation

Next, we provide a further justification of and insight into the dynamics model by introducing a game-theoretic formulation that is equivalent to Eqs. (4) and (5).

Consider a $(K + L)$-player game involving $K$ viewer groups and $L$ provider groups. Each viewer group selects a pure strategy $\lambda_k \in \mathbb{R}_{\geq 0}$, and each provider group chooses a pure strategy $\lambda_l \in \mathbb{R}_{\geq 0}$. The utility functions for the viewer and provider groups, denoted by $\{u_k\}_{k=1}^{K}$ and $\{v_l\}_{l=1}^{L}$ are defined as follows:

$$u_k(\boldsymbol{\lambda}) = \lambda_k \cdot \bar{\lambda}_k \left( \sum_{l=1}^{L} \pi_{k,l} \left( b_{k,l} + f_{k,l}(\lambda_l) \right) \right) - \frac{\lambda_k^2}{2}, \quad (6)$$

$$v_l(\boldsymbol{\lambda}) = \lambda_l \cdot \bar{\lambda}_l \left( \sum_{k=1}^{K} \pi_{k,l} \lambda_k \right) - \frac{\lambda_l^2}{2}, \quad (7)$$

We denote this game as $\mathcal{G}(\boldsymbol{\pi}, B, f, \bar{\lambda})$, where $B$ is a $K$-by-$L$ matrix whose $(k, l)$-element is $b_{k,l}$. Observation 1

establishes a connection between the game instance $\mathcal{G}$ and the dynamical formulation presented in Section 2.1.

**Observation 1.** *If all players in $\mathcal{G}$ apply gradient ascent to optimize their utility functions with learning rates $\{\eta_k\}_{k=1}^{K}$ and $\{\eta_l\}_{l=1}^{L}$, the resulting joint strategy evolving dynamics are exactly given by Eqs. (4) and (5).*

Observation 1 provides a first-principles perspective for understanding the dynamical formulation in Eqs. (4) and (5).[2] That is, we can interpret $\bar{\lambda}(\cdot)$ as the marginal gain from increasing the size of a viewer or provider group by one unit. Consequently, the first terms $\lambda_k \cdot \bar{\lambda}_k(\cdot)$ and $\lambda_l \cdot \bar{\lambda}_l(\cdot)$ represent the collective payoffs for viewer and provider groups of sizes $\lambda_k$ and $\lambda_l$. The quadratic terms $-\frac{\lambda_k^2}{2}$ and $-\frac{\lambda_l^2}{2}$ capture the congestion costs associated with maintaining larger populations (e.g., if a provider group becomes too large, providers within the group may face intensified competition and thus reduce their productivity due to diminished marginal gains). This suggests that Eqs. (4) and (5) are quite reasonable formulation to capture real-world interactions.

## 3. Stability and sub-optimality

This section provides theoretical analyses[3] of the stability and sub-optimality under the two-sided dynamics.

### 3.1. Stability

An important question to ask about the two-sided dynamics is on stability: ***Under what conditions do the dynamics converge to a fixed point?*** The following Theorem 4 provides an affirmative answer, demonstrating that the two-sided dynamics always converge to a stable fixed point, which is also a Nash Equilibrium (NE) (Nash Jr, 1950) of the corresponding game instance.[4]

**Theorem 1.** *For any continuous functions $f, \bar{\lambda}$ with bounded first-order derivatives, consider the environment defined by the game instance $\mathcal{G}(\boldsymbol{\pi}, B, f, \bar{\lambda})$. We have:*

1. *The NE of $\mathcal{G}$ always **exists**, but is not necessarily unique;*
2. *The two-sided dynamics (Eqs. (4) and (5)) always **converge** to one of $\mathcal{G}$'s NE, provided that $\eta_k, \eta_l$ are smaller than a constant that depends on the game parameters.*

Theorem 1 establishes a general stability result for the two-sided dynamics, showing that as long as the reactiveness hyperparams are sufficiently small, it always converges to

some fixed point corresponding to an NE of $\mathcal{G}$. This result is surprising in two aspects. First, it stands in stark contrast to prior work on one-sided markets, where NEs can fail to exist (Hron et al., 2022; Jagadeesan et al., 2022; Yao et al., 2023a). In comparison, Theorem 1 demonstrates that when the previously passive resource (e.g., viewer attention) becomes an active participant—as in our two-sided market model—the resulting market dynamics always admit a stable equilibrium. Second, it is well-documented that the existence of NEs does not guarantee convergence to them under gradient-based dynamics, as such dynamics often get stuck in local equilibria (Yao et al., 2024b) or even converge to non-Nash stationary points (Mazumdar et al., 1901). In contrast, we establish that gradient-based dynamics—specifically in our setting, the two-sided dynamics—provably converge to an NE under mild conditions. This highlights a nice structural property of the two-sided market.

Moreover, our result accommodates a wide range of modeling choices, including reference functions $\bar{\lambda}$, population effects $f$, and recommendation policies $\boldsymbol{\pi}$, without requiring restrictive assumptions. The following sufficient condition indicates an interesting relationship between the policy design and the stability of fixed points.

**Proposition 1** (Sufficient condition for stability). *Suppose that the first-order derivative of dynamics functions are bounded as $(\nabla_{e_l} \bar{\lambda}_l)(\nabla_{\lambda_l} f_l) \leq C_1, \forall l \in [L]$ and $(\nabla_{s_k} \bar{\lambda}_k) \leq C_2, \forall k \in [K]$ at some fixed point $\boldsymbol{\lambda}_{eq}$. Also, suppose $\eta_k \leq \eta, \forall k \in [K]$. Then, $\boldsymbol{\lambda}_{eq}$ is stable when*

$$\sum_{k=1}^{K} \pi_{l,k} \leq \frac{4\eta^{-1}}{C_1 C_2}. \tag{8}$$

Consider an *exposure-fair* policy, which distributes the exposure equally among provider subgroups. When $f$ and $\bar{\lambda}$ are monotonically increasing concave functions, Proposition 1 suggests that such an exposure-fair policy guarantees a balanced equilibrium, where both viewer and provider subgroups maintain a moderate population and payoffs without polarization. This is due to the following reasons. First, the upper bound (i.e., RHS of the inequality) becomes more restrictive when the first order derivative of the dynamics (i.e., $C_1$ and $C_2$) is large, which is true when viewer satisfaction ($s_k$), provider exposure ($e_l$), and provider population ($\lambda_l$) are small. Then, an exposure-fair policy can exclude such equilibria due to the violation of Ineq. (8). In contrast, using an *exposure-concentrated policy*, which does not distribute exposure to some provider subgroups, can lead to a polarized equilibrium with winners and losers, as such equilibria are not excluded by Ineq. (8).

Consequently, the reduced subgroup population may negatively impact the long-term viewer satisfaction, as we have seen in Figure 1. We formally discuss such impacts through the regret analysis in the next subsection.

---

[2]The game $\mathcal{G}$ resembles the Cournot Duopoly competition (Cournot, 1838). When $K = L = 1$ and $\bar{\lambda}(\mu) = a - b\mu$ and $\bar{\mu}(\lambda) = a - b\lambda$ for some positive constants $a$ and $b$, the game $\mathcal{G}$ corresponds exactly to the Cournot Duopoly model. The key distinction in ours is that $\bar{\mu}$ and $\bar{\lambda}$ are generic increasing functions.

[3]All proofs are provided in Appendix B.

[4]Definitions 1-3 in the Appendix formally define the concepts of fixed point, stability, and Nash equilibrium.

## 3.2. Sub-optimality

Our next question is: ***How does the "population effect" affect the policy design when the dynamics converge?*** To answer the question, we introduce the following notion of sub-optimality, called *regret*, to measure the performance difference between the optimal (static) policy[5] $\boldsymbol{\pi}^*$ and a given (possibly time-varying) policy $\boldsymbol{\pi}_t$:

$$\text{Regret}(\boldsymbol{\pi}) = \frac{1}{T}\sum_{t=1}^{T}\left(R(\boldsymbol{\pi}^*;\boldsymbol{\lambda}_t^*) - R(\boldsymbol{\pi}_t;\boldsymbol{\lambda}_t)\right)$$

where $\boldsymbol{\lambda}_t^*$ is the population at timestep $t$ under the policy $\boldsymbol{\pi}^*$ and $\boldsymbol{\lambda}_t$ is that of $\boldsymbol{\pi}$. $T$ is the total horizon of the timesteps. Assuming that the policy $\boldsymbol{\pi}_t$ converges to within $\delta$ of a static policy $\boldsymbol{\pi}$, the above regret can be decomposed into two factors as shown in the following Proposition 2.

**Theorem 2** (Regret decomposition). *The (total) regret is decomposed into two main factors:*

$$\text{Regret}(\boldsymbol{\pi}) = \underbrace{\frac{1}{T}\sum_{t=1}^{T}\Delta R(\boldsymbol{\lambda}_t^*,\boldsymbol{\lambda}_t^{\boldsymbol{\pi}})}_{(1)} + \underbrace{\frac{1}{T}\sum_{t=1}^{T}\Delta R(\boldsymbol{\pi}_t^1,\boldsymbol{\pi}_t)}_{(2)}$$

$$+ \mathcal{O}\left(\delta/T\right) + const.$$

*We call (1) as "population regret" and (2) as "policy regret". Each component is defined as follows.*

$$\Delta R(\boldsymbol{\lambda}_t^*,\boldsymbol{\lambda}_t) := R(\boldsymbol{\pi}_t^{1,*};\boldsymbol{\lambda}_t^*) - R(\boldsymbol{\pi}_t^1;\boldsymbol{\lambda}_t)$$
$$\Delta R(\boldsymbol{\pi}_t^1,\boldsymbol{\pi}_t) := R(\boldsymbol{\pi}_t^1;\boldsymbol{\lambda}_t) - R(\boldsymbol{\pi}_t;\boldsymbol{\lambda}_t),$$

*where $\boldsymbol{\pi}_t^1$ is the one-step myopic-greedy policy at timestep $t$ given population $\boldsymbol{\lambda}_t$, and $\boldsymbol{\pi}_t^{1,*}$ is that of under $\boldsymbol{\lambda}_t^*$.*

The population regret refers to the sub-optimality caused by the difference of population ($\boldsymbol{\lambda}_t$ and $\boldsymbol{\lambda}_t^*$) at timestep $t$, while the policy regret refers to the sub-optimality caused by the difference of the policy ($\boldsymbol{\pi}_t$ and $\boldsymbol{\pi}_t^1$). This suggests that the myopic policy makes the policy regret small, but completely ignores the population regret. This presents the reason why we have observed that the uniform random policy outperformed the myopic policy in the toy example presented in Figure 1 in the Introduction.

## 4. When is "myopic-greedy" optimal?

We have seen that the myopic-greedy policy is not always optimal. Then, the next question will be as follows: ***When does the myopic-greedy policy succeed?*** This section answers the question with a game-theoretic analysis in the case that $f$ and $\bar{\lambda}$ are linear functions.

Our main finding is that a myopic-greedy policy is nearly optimal when the provider population effects $f$ are homogeneous across provider groups. To formalize this, we define

the family of $\epsilon$-greedy policies as follows:

$$\pi_{k,l}^{(\epsilon)} = (1-\epsilon)\,\mathbb{I}\{l = \arg\max_{l'\in[L]}b_{k,l'}\} + \epsilon/L,$$

where $\mathbb{I}\{\cdot\}$ is the indicator function and $\epsilon \in [0,1]$ is the degree of exploration (i.e., random choice). Notably, $\boldsymbol{\pi}^{(0)}$ corresponds to the myopic-greedy policy, while $\boldsymbol{\pi}^{(1)}$ is the uniform random. The subsequent results establish that $\boldsymbol{\pi}^{(0)}$ optimal in the homogeneous-linear setting.

**Theorem 3** (Optimality of the myopic-greedy). *Let $\boldsymbol{\lambda}_\infty$ be the population at the NE under policy $\boldsymbol{\pi}$. For any base utility $B$ and linear increasing and homogeneous functions $\bar{\lambda}$ and $f$, the social welfare $R(\boldsymbol{\pi}^{(\epsilon)};\boldsymbol{\lambda}_\infty^{(\epsilon)})$ under the $\epsilon$-greedy policy is decreasing in $\epsilon \in [0,1]$. In particular, we have*

1. *When $K = 1$, $R(\boldsymbol{\pi}^{(\epsilon)};\boldsymbol{\lambda}_\infty^{(\epsilon)})$ is strictly decreasing in $\epsilon$.*
2. *When $K > 1$, we can identify functions $g, h$ such that*

$$g(\epsilon) \le R(\boldsymbol{\pi}^{(\epsilon)};\boldsymbol{\lambda}_\infty^{(\epsilon)}) \le g(\epsilon)h(\epsilon), \qquad (9)$$

*and both $g, h$ are decreasing in $\epsilon$. In addition, when $(\nabla_{\lambda_l}f_{k,l})(\nabla_{e_l}\bar{\lambda}_l)(\nabla_{s_k}\bar{\lambda}_k)$ is sufficiently small, the function $h(\epsilon) \to 1$ and Eq. (9) is tight.*

Theorem 3 suggests that when the population effect $f$ is linear and homogeneous across different provider groups, the myopic-greedy policy will be always optimal. This also holds in the case when there is no population effect, i.e., $f_{k,l}(\lambda_l)$ always equals to a constant, $\forall \lambda_l \in \mathbb{R}, \forall (k,l) \in [K] \times [L]$. In such cases, the use of the myopic-greedy policy is recommended.

However, when the population effect becomes heterogeneous across different provider groups, the myopic policy ceases to be optimal, as illustrated by Proposition 2.

**Proposition 2.** *The myopic-greedy policy can be sub-optimal when $\{f_{k,l}\}$ are heterogeneous across provider groups, even when $\bar{\lambda}$ and $f$ remain linear.*

We provide a detailed example in Appendix B.5 to support Proposition 2. Intuitively, the heterogeneity matters because it results in *cross-over* behaviors (e.g., provider group A starts low utility but becomes high utility, while provider group B has medium utility regardless of the population), which matter for policy design. Aside from the linear case, *saturation* behaviors (e.g., no further increases in utility once the population becomes adequately large) also matter. When we encounter such heterogeneous or concave population effects among multiple provider groups, the myopic-greedy policy may not be optimal, as it ignores the impact of policy to future population changes. These results demonstrate that the myopic-greedy policy is guaranteed optimal only under highly restrictive conditions and emphasize the need for practical solutions to account for the long-term effects.

# 5. Optimizing the long-term social welfare

The key observation from the previous sections is that myopic-greedy policy fails by ignoring the *population regret*, which comes from the difference between the population of the current policy and that of the optimal one. Therefore, we first establish a policy learning method that optimizes for the population regret and later consider balancing this with the policy regret. However, one difficulty in minimizing the population regret $\Delta R(\boldsymbol{\lambda}_t^*, \boldsymbol{\lambda}_t)$ (Theorem 2) is that the population $\boldsymbol{\lambda}_t$ depends on the past choices of policy $\pi$. This means that when optimizing the policy, we should take into account its future influence on the population. Because we know that the population gradually changes towards the reference population $\bar{\lambda}(\cdot)$, we consider the following **Look-ahead policy**:

$$\boldsymbol{\pi}_t^{(d)} := \arg\max_{\boldsymbol{\pi} \in \Pi} R(\bar{\boldsymbol{\pi}}_t^1(\boldsymbol{\pi}); \bar{\boldsymbol{\lambda}}_t(\boldsymbol{\pi})) \qquad (10)$$

Above, $\bar{\boldsymbol{\lambda}}_t(\boldsymbol{\pi})$ is the reference population at timestep $t$ given the viewer satisfaction and provider exposure realized by the policy $\boldsymbol{\pi}$ at population $\boldsymbol{\lambda}_t$, i.e., $\bar{\lambda}_k(s_{t,k})$ and $\bar{\lambda}_l(e_{t,l})$. $\bar{\boldsymbol{\pi}}_t^1(\boldsymbol{\pi})$ is the myopic-greedy policy at the reference population $\bar{\boldsymbol{\lambda}}_t(\boldsymbol{\pi})$. Thus, the lookahead policy focuses on reaching reference populations which enable high user satisfaction.

The look-ahead policy's optimization problem is potentially nonconvex. To make it differential, one can consider the following softmax policy as the approximation of $\bar{\boldsymbol{\pi}}_t^1$:

$$\bar{\pi}_{t,k,l}^1 = \frac{\exp(\gamma \cdot (b_{k,l} + f_{k,l}(\bar{\lambda}_l(e_l))))}{\sum_{l' \in [L]} \exp(\gamma \cdot (b_{k,l'} + f_{k,l'}(\bar{\lambda}_{l'}(e_{l'}))))}$$

where $\gamma > 0$ is the inverse temperature parameter. Then, we can optimize the objective function in Eq. (10) via gradient ascent, where we present the exact gradient in Appendix A.

Once we obtain the look-ahead policy, we can interpolate between the look-ahead policy and the myopic-greedy policy to balance the population and policy regrets as follows:

$$\boldsymbol{\pi}_t = \beta \boldsymbol{\pi}_t^{(d)} + (1 - \beta) \boldsymbol{\pi}_t^{(m)} \qquad (11)$$

where $\beta \in [0, 1]$ is the interpolation hyperparameter and $\boldsymbol{\pi}_t^{(m)}$ is the myopic-greedy policy. $\beta$ can be determined by the platform's desire to focus on short vs. long term goals, while we later show that $\beta = 1.0$ can be a reasonable choice from the experiment results.

## 5.1. Estimation the dynamics

In practice, there may be situations in which we need to estimate the dynamics function ($\bar{\lambda}$ and $f$) using some function approximation. In such case, we can use the following Explore-then-Commit style estimation:

1. For $t \leq T_b$, deploy some epsilon-greedy policy and collect the data of $D_t := (s_k, e_l, q_{k,l}, \lambda_k, \lambda_l), \forall (k, l) \in$

$[K] \times [L]$. Then, update the dataset as $\mathcal{D}_t = \mathcal{D}_{t-1} \cup D_t$ where $\mathcal{D}_0 := \phi$ (empty set). $T_b$ is a burn-in period.
2. For $t > T_b$, update the dynamics models $(\widehat{q}_{l,k}(\lambda_l), \widehat{\lambda}(s_k), \widehat{\lambda}(e_l))$ via supervised learning. Note that the true "reference population" is obtained from data as $\bar{\lambda}_{t'} = \eta^{-1}(\lambda_{t'+1} - \lambda_{t'}) + \lambda_{t'}$ from Eqs. (4) and (5). Then, optimize the policy with Eqs. (10) and (11) using the estimated functions $\widehat{\lambda}$ and $\widehat{q}$, collect the interaction data $D_t$, and add them to the dataset $\mathcal{D}_t$.

# 6. Synthetic Experiment

We first study the dynamics and the performance of the proposed method in a synthetic experiment. In this task, we use $K = L = 20$ subgroups. To define the base utility, we first sample 20-dimensional binary feature vectors $(u_k, c_l)$ from a Bernoulli distribution for each viewer and provider group and let their inner products be the base utility $b_{k,l} = u_k^\top c_l, \forall (k, l) \in [K] \times [L]$. Then we simulate the following concave dynamics:

$$\bar{\lambda}_k(z) = \lambda_k^{(max)}(\sigma(z/\tau_k^{(\lambda)}) - 0.5), \qquad (12)$$

where $\sigma(z) := 1/(1 + \exp(-z))$ is the sigmoid function, and $\bar{\lambda}(\cdot)$ follows the upper half of the sigmoid function.

Next, to simulate a *heterogeneous* population effect, we further take inner products between viewer embeddings $u_k$ and the vector of population-dependent quality as follows.

$$f_{k,l}(\lambda_l) = u_k^\top [\bar{f}_l^{(1)}(\lambda_l), \bar{f}_l^{(2)}(\lambda_l), \cdots, \bar{f}_l^{(d)}(\lambda_l)], \quad (13)$$

where its $i$-th quality element follows the upper half of the sigmoid, i.e., $\bar{f}_l^{(i)}(z) = F_l^{(i),(max)}(\sigma(z/\tau_l^{(i),(F)}) - 0.5)$. We use $d = 20$. With this model, each provider group has different improvements in quality of content, e.g., visuals, humor, and technical depth, and each viewer group has different preferences on these aspects of quality. We visualize the population effect in Figure 6 in the Appendix.

We initialized the subgroups populations by sampling values from the normal distribution, so we have the majority and minority subgroups at $t = 0$. Specifically, we use two initializations: (1) a small population ($\lambda \sim \mathcal{N}(20, 10^2)$) and (2) a large one ($\lambda \sim \mathcal{N}(100, 30^2)$) to see how policies perform in both increasing and decreasing dynamics.

**Compared methods**. We compare the proposed look-ahead policy with varying interpolation hyperparameter $\beta \in [0.0, 0.2, \ldots, 1.0]$. We also compare with the uniform random policy as a reference. When computing the look-ahead policy, we assume access to the dynamics and population effect functions. The lookahead policy is computed with gradient ascent on the objective (Eq. (10)) for 100 iterations.

**Results.** We run the compared methods for 200 timesteps and report the results in Figure 2. The results demonstrate

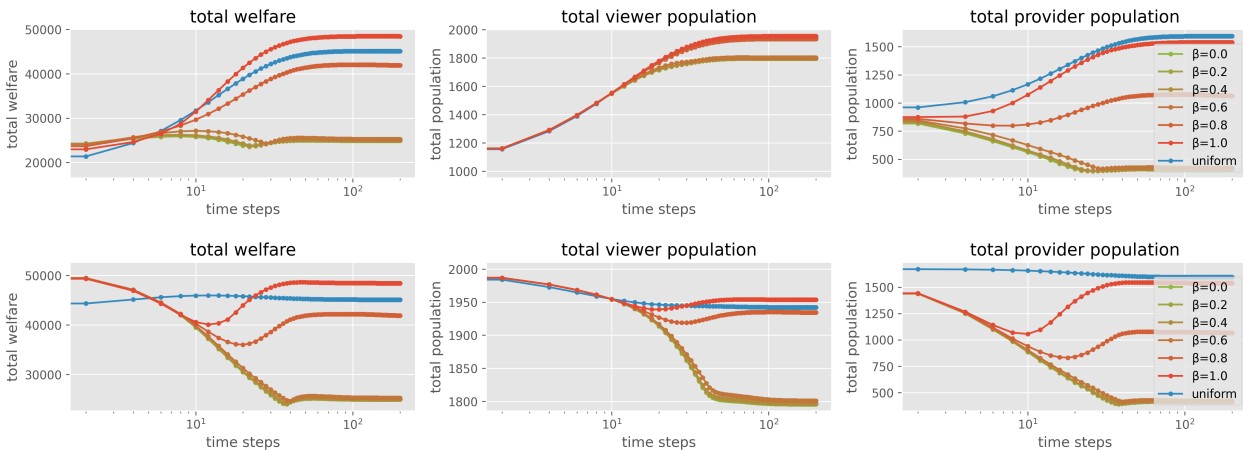

*Figure 2.* **Comparing the total welfare, and the viewer and provider populations with varying values of interpolation hyperparam, i.e., $\beta$. (Top)** small initial population and **(Bottom)** large initial population. "uniform" represents the uniform random policy.

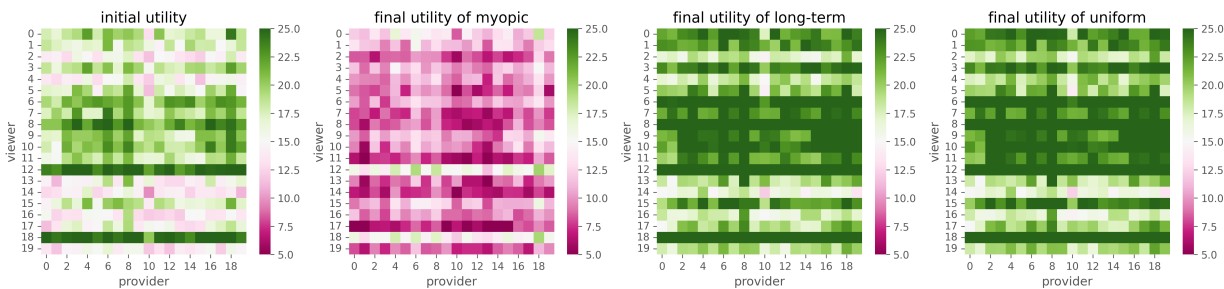

*Figure 3.* **Comparing the utility matrix of the myopic ($\beta = 0.0$), long-term ($\beta = 1.0$), and uniform random policies at the final timestep and the initial utility matrix.** For the initial utility matrix, we use the one with a small initial population.

that the long-term (look-ahead) policy performs better than the myopic-greedy policy, as the reward gain from the population effects is large in this setting. Specifically, we observe that the pure look-ahead policy ($\beta = 1.0$) increases the provider populations while the myopic-greedy policy decreases the provider populations. These population changes immediately affect the total welfare, suggesting that guaranteeing high population among multiple subgroups via balanced exposure allocation is crucial when population effects matter. Indeed, we also observe a different distribution of the utility matrix at the final timestep across compared methods in Figure 3. Interestingly, while the uniform policy has the largest provider population at the final timestep in Figure 2, the look-ahead policy ($\beta = 1.0$) achieves better total welfare (and population regret). This is because the look ahead policy allocates exposure more efficiently than the uniform policy to ensure both high viewer satisfaction and high provider exposure among multiple subgroups, empirically demonstrating the effectiveness of our approach.

## 7. Real-data Experiment

This section studies the empirical behavior of the proposed method using the KuaiRec (Gao et al., 2022) dataset.

**Datasets.** KuaiRec (dense) (Gao et al., 2022) is a viewer-provider interaction dataset consisting of 4,676,570 data samples with 1,411 viewers and 3,326 videos (i.e., providers). The data contains "watch ratio" (i.e., play duration divided by the video duration) as the viewers feedback signal. We clip the maximum watch ratio by 10 and learn the viewer-provider base utility $b(u, c)$ using a neural collaborative filtering (CF) model (He et al., 2017). The base utility is calculated as individual level ($(u, c) \in \mathcal{U} \times \mathcal{C}$), where $u$ and $c$ are viewer and provider embeddings.

**Simulation.** We simulate the subgroup of viewers and providers, following the procedure presented in Bose et al. (2023). Specifically, we cluster viewers and providers into $K = L = 20$ subgroups respectively, based on the viewer and provider embeddings learned by the neural CF model (He et al., 2017). We use the same initialization and dynamics of the population as described in Section 6. Then,

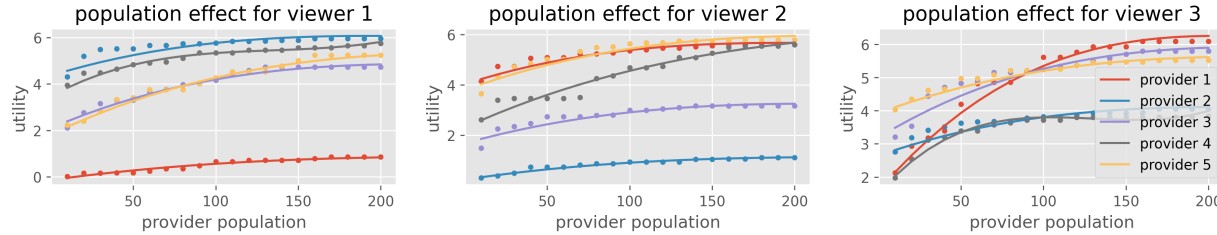

*Figure 4.* **Visualization of the (true) population effects in the real-world experiment.** The population effects are based on the spline function (Reinsch, 1967) fitted on the empirical population effect (dotted points) observed in the KuaiRec (Gao et al., 2022) dataset. Figures 7 and 8 in the Appendix also report the population effects and dynamics learned by the long-term policy, following Section 5.1.

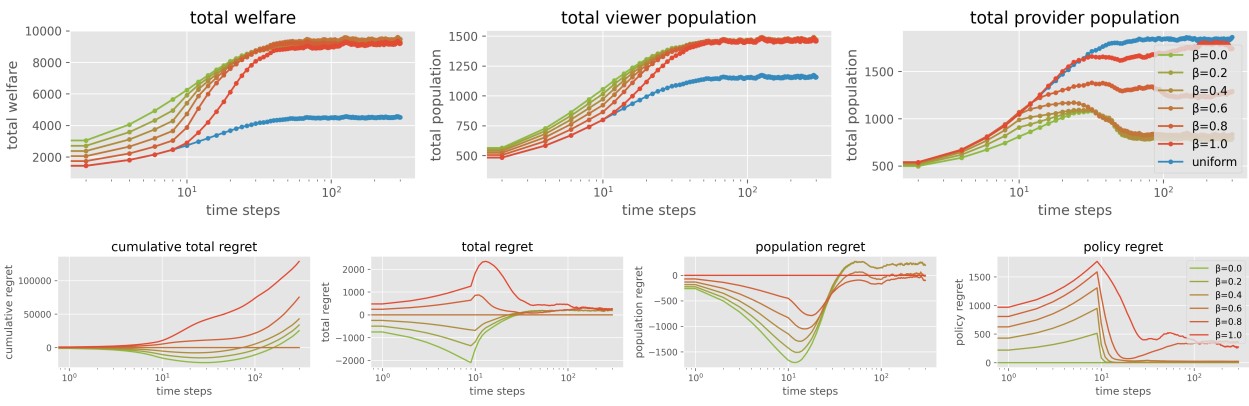

*Figure 5.* **Comparing the total welfare, viewer and provider populations, and regrets in the real-data experiment.** Cumulative regret is the sum of total regret by the timestep $t$, and the total regret is decomposed into the population and policy regrets. Note that the true optimal policies that minimize the total regret and population regret are not accessible. Thus, we report the empirical regrets by letting one of the compared policies as the optimal baseline.

we simulate the utility and population effects as follows.

1. Let $u_k := \mathbb{E}[u \mid u \in \mathcal{U}_k]$ be the mean embeddings of viewer group $k$. We define the group-wise base utility as $b_{l,k} = \mathbb{E}[b(u_k, c) \mid c \in \mathcal{C}_l]$ (i.e., mean utility that viewer group $k$ receives from providers in group $l$).

2. Next, to simulate a population effect, we generate a random permutation of providers within each provider group. Given the current provider population $\lambda_{t,l}$, we let first $\lambda_{t,l}$ samples in the permutation as the the set of providers in the subgroup $l$ used at timestep $t$. We denote this subset as $\mathcal{C}_{t,l}$. Then, we define the utility from the provider group $l$ as $q(u_k, c_l) := \arg\max_{c \in \mathcal{C}_{t,l}} b(u_k, c)$. Therefore, the population effects are defined as

$$f_{k,l}(\lambda_l) := \arg\max_{c \in \mathcal{C}_{t,l}(\lambda_{t,l})} b(u_k, c) - b_{k,l},$$

which increases monotonically as provider population ($\lambda_{t,l}$) increases.

To obtain a smooth population effect function, we generate 10 different random permutation in Step 2. Then, we take the average of 10 population effects and fit spline func-

tions (Reinsch, 1967) implemented in SciPy (Virtanen et al., 2020). The resulting population effects are in Figure 4.

**Estimation of the dynamics.** In this experiment, we estimate the dynamics functions $\bar{\lambda}$ and $f$ using regression. We use the model $F(z) = a_0(1-\exp(-a_1(x-a_2)))+a_3$ due to its concavity and flexibility, and fit the params $(a_0, \cdots, a_3)$ from interaction data as described in Section 5.1. Note that we add perturbations in the population dynamics $\xi_t$ sampled from a normal distribution as $\xi_t \sim \mathcal{N}(0.0, 0.01^2) \times \boldsymbol{\lambda}_t$ (i.e., the scale of perturbation is proportional to the population) to account for the difficulty in learning the real-world dynamics. During the burn-in period (10 steps), we deploy epsilon-greedy with the corresponding value of $\beta (= \epsilon)$.

**Results.** Figure 5 report the population dynamics, total welfare, and the regret. Unlike synthetic experiment, the myopic-greedy policy performs better than the uniform random policy, while the proposed look-ahead policy is competitive to the myopic-policy in the total welfare after converging to the NE. However, we observe some tradeoff between the myopic-greedy and look-ahead policy. Specifically, Figure 5 (Bottom) suggests that the myopic-greedy

policy ($\beta = 0.0$) has some population regret compared to the look-ahead policy ($\beta = 1.0$), while the look-ahead policy retains some policy regret. As a result, an interpolated policy with $\beta = 0.6$ is the best among the compared methods, while all interpolated policies with various values of $\beta$ perform quite well. It is also worth mentioning that the look-ahead policy ($\beta = 1.0$) maintains high total welfare even while achieving a similar level of the provider population as the uniform policy. This suggests that the proposed look-ahead policy is able to allocate exposure efficiently by considering the long-term population effects.

Together with the synthetic experiment results, we observe that the proposed look-ahead policy adaptively behaves (near-)optimally in terms of total welfare, while also guaranteeing a high provider population through provider-fair exposure allocation. This minimizes the effort to tune the hyperparam $\beta$ as the look-ahead policy ($\beta = 1.0$) works reasonably well in practical situations.

## 8. Related work

**Population shifts.** The most relevant existing works to ours are Mladenov et al. (2020) and Huttenlocher et al. (2023), which consider the population dynamics by modeling the departure of viewers and providers. Specifically, Mladenov et al. (2020) assume that a provider will leave the platform if the provider cannot receive adequate exposure (i.e., exposure is below some given threshold). Then, Mladenov et al. (2020) solve the constrained optimization problem as a linear integer program and demonstrate that provider fairness is crucial to maintain a high viewer welfare. To extend, Huttenlocher et al. (2023) additionally consider the departure of viewers who receive less utility than given thresholds. Huttenlocher et al. (2023) also formulate a matching problem to determine which viewers and providers to keep in the platform to achieve high long-term social welfare. However, both works ignore the possible growth of the platform, and how a policy design affects the "growing-the-pie" behavior has remained underexplored (also, these existing works are not directly applicable to our problem setting). Our work complements these works by finding that provider fairness is important to ensure high "population effects" in a generalized setting.

Note that our work is also closely related to the performative prediction literature (Perdomo et al., 2020; Brown et al., 2022; Narang et al., 2023). This line of work studies the policy optimization under a stationary point of dynamics, when the algorithm affects the *states* (e.g., population) of the environment, and the optimal algorithm can change depending on the states. While these works (theoretically) discuss the model optimization and its convergence under a convex loss function (Brown et al., 2020), our work empirically demonstrates that the look-ahead policy gradient approach (i.e., non-convex optimization) can work in a complex and interactive dynamics of two-sided platforms.

**Strategic content providers.** Another related literature is the policy optimization under strategic content providers (Hron et al., 2022; Jagadeesan et al., 2022; Yao et al., 2023b). These works often formulate content providers as "selfish" agents who maximize only their own utility defined by the amount of exposure minus the cost of content generation. As described in Section 2.2, our problem setting can also be seen as a variant of policy optimization under strategic viewers and content providers. However, our formulation is distinctive in modeling the increase and decrease of the total population, while existing works assume that the total number of viewers and providers are fixed. This difference results in novel findings: while Hron et al. (2022) find that more exploration (randomness) can reduce incentives for producing high-quality "niche" content, we find that a more random provider-fair policy can be beneficial when taking the population growth of multiple groups into account.

**Fairness, diversity, and welfare.** Fairness and diversity among providers have been considered as necessary metrics or constraints when optimizing policies in two-sided platforms (Singh & Joachims, 2018; Wang & Joachims, 2021; Boutilier et al., 2023). While provider-fairness is initially considered important from provider-side perspectives (Singh & Joachims, 2018), recent works consider the impacts of provider fairness on viewer welfare. Specifically, provider fairness turned out important to maintain provider diversity (Yao et al., 2023b; Hron et al., 2022), and provider diversity helps maintain viewer welfare in the long-run (Su et al., 2023; Mladenov et al., 2020). Our findings align with these works in pointing out that provider-fairness is important for long-term viewer satisfaction, but from a different viewpoint based on the growth of populations.

## 9. Conclusion

This paper studies recommender policy design in two-sided platforms where viewer and provider populations matter. Through the control- and game-theoretic analyses, we found that the myopic-greedy policy is guaranteed to be optimal only when the population effects are linear and homogeneous among provider groups, and otherwise may fall short by ignoring the "population effects". To take such long-term effects into account, we proposed a simple algorithm and empirically demonstrate that it provides both viewer satisfaction and provider exposure for future population growth. We believe our work provides a cornerstone to build dynamics-aware allocation policies in two-sided platforms where multiple stakeholders engage.

## Acknowledgements

This work was partly funded by NSF CCF 2312774, NSF OAC-2311521, and a gift to the LinkedIn-Cornell Bowers CIS Strategic Partnership. Haruka Kiyohara is supported by the Funai Overseas Scholarship.

We thank Thorsten Joachims and Richa Rastogi for their thoughtful feedback on the early-stage discussion about the dynamics in two-sided platforms. We also thank anonymous reviewers for the valuable discussion during the review.

## Impact Statement

The real-world experiments are run on a public recommendation dataset, and we do not consider there to be any ethical concerns in our experiments.

For the broader social impact, this paper has shed light on the population effects in two-sided platforms for the first time. Our findings suggest that ensuring the provider exposure among multiple subgroups, instead of concentrating on a single subgroup, is important to maintain a high population growth and utility. This is also relevant to provider-side fairness (Singh & Joachims, 2018) and diversity (Su et al., 2023) in many decision-making scenarios (Extended related work in Appendix 8 provides a further discussion). Moreover, the population effects potentially matter in broad applications including recommender systems, social networking services, and even school admission or hiring processes. Working on such application-specific configurations would also be an impactful future direction of the research community.

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

## A. Derivation of the gradient of Eq. (10)

We derive the gradient of the look-ahead policy using the chain-rule as follows:

$$\nabla_{\boldsymbol{\pi}}(R(\bar{\pi}_t^1(\boldsymbol{\pi}); \bar{\boldsymbol{\lambda}}_t(\boldsymbol{\pi}))) = \nabla_{\boldsymbol{\pi}}\left(\sum_{k=1}^{K} \bar{\lambda}_k(\boldsymbol{\pi}) \sum_{l=1}^{L} \bar{\pi}_{k,l}^1(\boldsymbol{\pi})(b_{k,l} + f_{k,l}(\bar{\lambda}_l(\boldsymbol{\pi})))\right)$$

$$= \sum_{k=1}^{K}\sum_{l=1}^{L} \nabla_{\boldsymbol{\pi}}(\bar{\pi}_{k,l}^1)\,\bar{\lambda}_k(\boldsymbol{\pi})(b_{k,l} + f_{k,l}(\bar{\lambda}_l(\boldsymbol{\pi})))$$

$$+ \sum_{k=1}^{K} \nabla_{\boldsymbol{\pi}}(\bar{\lambda}_k) \sum_{l=1}^{L} \bar{\pi}_{k,l}^1(\boldsymbol{\pi})(b_{k,l} + f_{k,l}(\bar{\lambda}_l(\boldsymbol{\pi})))$$

$$+ \sum_{k=1}^{K} \bar{\lambda}_k(\boldsymbol{\pi}) \sum_{l=1}^{L} \nabla_{\bar{\lambda}_l}(f_{k,l})\nabla_{\boldsymbol{\pi}}(\bar{\lambda}_l)\,\bar{\pi}_{k,l}^1(\boldsymbol{\pi}),$$

where the $(k', l')$-th element of the gradient matrix is defined as follows:

$$\nabla_{\boldsymbol{\pi}}(\bar{\lambda}_k)_{k',l'} = \nabla_{s_k}(\bar{\lambda}_k)\nabla_{\boldsymbol{\pi}}(s_k)_{k',l'} = \nabla_{s_k}(\bar{\lambda}_k) \cdot \mathbb{I}\{k' = k\}\, q_{k',l'},$$
$$\nabla_{\boldsymbol{\pi}}(\bar{\lambda}_l)_{k',l'} = \nabla_{e_l}(\bar{\lambda}_l)\nabla_{\boldsymbol{\pi}}(e_l)_{k',l'} = \nabla_{e_l}(\bar{\lambda}_l) \cdot \mathbb{I}\{l' = l\}\lambda_{k'}.$$

Note that $\nabla_{\bar{\lambda}_l}(f_{k,l})$, $\nabla_{s_k}(\bar{\lambda}_k)$, and $\nabla_{e_l}(\bar{\lambda}_l)$ can be the gradient of the estimated dynamics and population effect functions, when the true dynamics are not accessible (i.e., we can use the estimation process described in Section 5.1).

When implementing the algorithm, one can also use `autograd` implemented in PyTorch (Paszke et al., 2019) to calculate the gradient directly from the look-ahead objective.

## B. Omitted Proofs

This section provides proofs for the Theorems and Propositions presented in the main text. Note that we define the fixed point, stability, and Nash equilibrium as follows.

**Definition 1** (Fixed point). *Let $S(\cdot)$ be the dynamics function that maps the population from the previous timestep to the next timestep, i.e., $\lambda_{t+1} = S(\lambda_t)$. Then, $\boldsymbol{\lambda}_{eq}$ is a fixed point under the policy $\boldsymbol{\pi}$ when $\boldsymbol{\lambda}_{eq}$ satisfies,*

$$\boldsymbol{\lambda}_{eq} = S(\boldsymbol{\lambda}_{eq}),$$

*where $S$ is a static function under a static policy $\boldsymbol{\pi}$.*

**Definition 2** (Stability). *A fixed point $\boldsymbol{\lambda}_{eq}$ is stable under the policy $\boldsymbol{\pi}$ when $\boldsymbol{\lambda}_{eq}$ satisfies,*

$$\forall \epsilon > 0, \exists \delta > 0, \quad \|\boldsymbol{\lambda}_0 - \boldsymbol{\lambda}_{eq}\| < \delta \implies \|\boldsymbol{\lambda}_t - \boldsymbol{\lambda}_{eq}\| < \epsilon.$$

*This condition is satisfied when $\det(\nabla_{\lambda} S) < 1$ holds.*

**Definition 3** (Nash equilibrium). *$\boldsymbol{\lambda}^*$ is a Nash equilibrium of game $\mathcal{G}(\boldsymbol{\pi}, B, f, \bar{\lambda})$ under the policy $\boldsymbol{\pi}$ when $\boldsymbol{\lambda}^*$ satisfies,*

$$\lambda_k^* = \arg\max_{\lambda_k}\, u_k(\lambda_k, \boldsymbol{\lambda}_{-k} = \boldsymbol{\lambda}_{-k}^*),$$
$$\lambda_l^* = \arg\max_{\lambda_l}\, v_l(\lambda_l, \boldsymbol{\lambda}_{-l} = \boldsymbol{\lambda}_{-l}^*),$$

*where $\boldsymbol{\lambda}_{-k}, \boldsymbol{\lambda}_{-l}$ denotes the vectors that contain all elements of $\boldsymbol{\lambda}^*$ except $\lambda_k$ and $\lambda_l$, and $u_k, v_l$ are determined by Eqs. (6) and (7).*

Based on the definition of fixed point, we prove the properties of the policy and the corresponding fixed point below.

### B.1. Proof of Proposition 1

To prove Proposition 1, we first introduce the following Theorem 4. We also use the fixed point described in the following in the proof of Theorem 1.

**Theorem 4** (Conditions for a fixed point). $\boldsymbol{\lambda}_{eq}$ *is a stable equilibrium when the following is satisfied for all $l \in [L]$.*

$$\eta_l(1 - \eta_l)(\nabla_{e_l}\bar{\lambda}_l)(\nabla_{\lambda_l}f_l)\sum_{k=1}^{K}\eta_k(\nabla_{s_k}\bar{\lambda}_k)\pi_{k,l} < 1$$

*where $(\nabla\cdot)$ is the first-order derivative at $\boldsymbol{\lambda}_{eq}$.*

*Proof.* We prove the condition for the stable equilibrium. Let $S(\cdot)$ be the dynamics function that maps the population from the previous timestep to the next timestep, i.e., $\lambda_{t+1} = S(\lambda_t)$. Then we have

$$\lambda_{t+1} - \lambda_{eq} = (\nabla_\lambda S)(\lambda_t - \lambda_{eq}).$$

This matrix is decomposed into four sub-matrices as

$$(\nabla_\lambda S) = \begin{bmatrix} A_{1,1} & A_{1,2} \\ A_{2,1} & A_{2,2} \end{bmatrix}$$

where $A_{1,1}$ is a $(K, K)$-dimensional matrix, $A_{1,2}$ is a $(K, L)$-dimensional matrix, $A_{2,1}$ is a $(L, K)$-dimensional matrix, and $A_{2,2}$ is a $(L, L)$-dimensional matrix. From the dynamics equations 4 and 5, the element of each matrix is derived as

$$\{A_{1,1}\}_{k,k'} = (\nabla_{\lambda_{k'}}\lambda_k) = (1 - \eta_k)\mathbb{I}\{k = k'\}$$
$$\{A_{2,2}\}_{l,l'} = (\nabla_{\lambda_{l'}}\lambda_l) = (1 - \eta_l)\mathbb{I}\{l = l'\}$$
$$\{A_{1,2}\}_{k,l} = (\nabla_{\lambda_l}\lambda_k) = \eta_k(\nabla_{q_k}\bar{\lambda}_k)\pi_{k,l}(\nabla_{\lambda_l}f_l)$$
$$\{A_{2,1}\}_{l,k} = (\nabla_{\lambda_k}\lambda_l) = \eta_l(\nabla_{e_l}\bar{\lambda}_l)\pi_{k,l}$$

. When the spectrum radius (i.e., the maximum eigenvalues) of $(\nabla_\lambda S)$ is less than 1, $\lambda_{eq}$ is a stable equilibrium of the dynamics. Here, because $A_{1,1}$ and $A_{2,2}$ are invertible matrices, we can use the Schur complement as

$$\det(\nabla_\lambda S) = \det(A_{1,1})\det(\underbrace{A_{2,2} - A_{2,1}A_{1,1}^{-1}A_{1,2}}_{:=A'}).$$

Therefore, when the eigenvalues of $A_{1,1}$ is $\mu_1$ and that of $A'$ is $\mu_2$, the eigenvalues of $(\nabla_\lambda S)$ is $\mu = [\mu_1, \mu_2]$. Thus, the eigenvalues of $(\nabla_\lambda S)$ are

$$\{\mu_1\}_k = (1 - \eta_k)$$

$$\{\mu_2\}_l = \eta_l(1 - \eta_l)(\nabla_{e_l}\bar{\lambda}_l)(\nabla_{\lambda_l}f_l)\sum_{k=1}^{K}\eta_k(\nabla_{q_k}\bar{\lambda}_k)\pi_{k,l}$$

$\square$

When $\eta_k \leq \eta, \forall k \in [K]$ hold and the gradient norm is bounded as $(\nabla_{e_l}\bar{\lambda}_l)(\nabla_{\lambda_l}f_l) \leq C_1$ and $(\nabla_{q_k}\bar{\lambda}_k) \leq C_2$ at $\lambda_{eq}$, we have Proposition 1:

$$\sum_{k=1}^{K}\pi_{k,l} \leq \frac{4\eta^{-1}}{C_1 C_2},$$

where we use $\forall \eta_l \in [0, 1), \eta_l(1 - \eta_l) \leq 0.25$.

## B.2. Proof of Theorem 1

*Proof.* **Existence of NE:** First we show the Nash equilibrium must exist. Note that $u_k, v_l$ take negative values as $\lambda_k, \lambda_l$ become sufficiently large, we can without loss of generality assume each player's strategy is upper bounded by a finite constant. As a result, for each player in the game, its strategy set is a convex, closed, and bounded region, and its utility

function is clearly concave in its own strategies. According to Theorem 1 in (Rosen, 1965), such a game is a concave $n$-person game and its Nash equilibrium must exist.

**Non-uniqueness of NE:** Next, we give an example showing that the Nash equilibrium is not necessarily unique, if we do not impose any assumption on $\bar{\lambda}_k, \bar{\lambda}_l, q$. Consider the case when $K = L = 1$ and the following configurations

$$\bar{\lambda}_k(\lambda^{(c)}) = \sigma(4(2\lambda^{(c)} - 1)), \quad \bar{\lambda}_l(\lambda^{(u)}) = \sigma(3(2\lambda^{(u)} - 1)),$$

where $\sigma(x) = 1/(1 + \exp(-x))$ is the sigmoid function. In this case, the two players have the following utility functions

$$u_1(\lambda^{(u)}, \lambda^{(c)}) = \lambda^{(u)} \cdot \sigma(4(2\lambda^{(c)} - 1)) - \frac{1}{2}(\lambda^{(u)})^2,$$

$$u_2(\lambda^{(c)}, \lambda^{(u)}) = \lambda^{(c)} \cdot \sigma(3(2\lambda^{(u)} - 1)) - \frac{1}{2}(\lambda^{(c)})^2. \tag{14}$$

Any fixed point of system (14) should satisfy the following first-order condition

$$\lambda^{(u)} = \sigma(4(2\lambda^{(c)} - 1)), \quad \lambda^{(c)} = \sigma(3(2\lambda^{(u)} - 1)), \tag{15}$$

and we can easily verify that Eq. (15) has three solutions

$$[\lambda^{(u)}, \lambda^{(c)}] = [0.0278, 0.0555], \quad [0.5, 0.5], \quad [0.9722, 0.9445]. \tag{16}$$

On the other hand, since each player's utility function is strictly concave in its own strategy, any fixed point of system (14) must correspond to a Nash equilibrium of the game. Hence, the game has three distinct Nash equilibria, which are given by Eq. (16).

**Convergence of two-sided dynamics:** According to Theorem 4, we know that two-sided dynamics converge to some stable fixed point $\boldsymbol{\lambda}_{eq}$, as long as for each reactiveness hyperparam $\eta$, it holds that

$$\sum_{k=1}^{K} \pi_{k,l} \leq \frac{4\eta^{-1}}{C_1 C_2}, \tag{17}$$

where $C_1$ and $C_2$ are upper bounds of $(\nabla_{e_l} \bar{\lambda}_l)(\nabla_{\lambda_l} f_l)$ and $(\nabla_{q_k} \bar{\lambda}_k)$ at $\boldsymbol{\lambda}_{eq}$. A sufficient condition for Eq. (17) to hold is $\eta \leq \frac{4}{KC_1C_2}$.

Next we argue that $\boldsymbol{\lambda}_{eq}$ must also correspond to the Nash equilibrium of $\mathcal{G}$ if $\eta \leq \frac{4}{KC_1C_2}$. This is because $\boldsymbol{\lambda}_{eq}$ being a stable point means that for each viewer $k$ and provider $l$ under $\boldsymbol{\lambda}_{eq}$, they cannot alter their strategies unilaterally to improve their payoffs $u_k$ or $v_l$ in a small region around $\boldsymbol{\lambda}_{eq}$. That is, $\lambda_k, \lambda_l$ given by $\boldsymbol{\lambda}_{eq}$ are the local maximum points of $u_k$ and $v_l$ under $\boldsymbol{\lambda}_{eq}$. Since both $u_k$ and $v_l$ are strictly concave quadratic functions in $\lambda_k$ and $\lambda_l$, their local maximum points must also be the global maximum points. Hence, they also cannot unilaterally improve their payoffs in their entire strategy sets. This demonstrates that $\boldsymbol{\lambda}_{eq}$ satisfies the definition of Nash equilibrium.

$\square$

### B.3. Proof of Theorem 2

*Proof.* Here, we provide a proof of the regret decomposition. First of all, we have

$$\begin{aligned}
&R(\boldsymbol{\pi}^*; \boldsymbol{\lambda}_t^*) - R(\boldsymbol{\pi}_t; \boldsymbol{\lambda}_t) \\
&= R(\boldsymbol{\pi}^*; \boldsymbol{\lambda}_t^*) - R(\boldsymbol{\pi}_t^{1,*}; \boldsymbol{\lambda}_t^*) + R(\boldsymbol{\pi}_t^{1,*}; \boldsymbol{\lambda}_t^*) - R(\boldsymbol{\pi}_t^1; \boldsymbol{\lambda}_t) + R(\boldsymbol{\pi}_t^1; \boldsymbol{\lambda}_t) - R(\boldsymbol{\pi}_t; \boldsymbol{\lambda}_t) \\
&= \Delta R(\boldsymbol{\lambda}_t^*, \boldsymbol{\lambda}_t) + \Delta R(\boldsymbol{\pi}_t^1, \boldsymbol{\pi}_t) + \text{const.}
\end{aligned}$$

where

- $\Delta R(\boldsymbol{\lambda}_t^*, \boldsymbol{\lambda}_t) := R(\boldsymbol{\pi}_t^{1,*}; \boldsymbol{\lambda}_t^*) - R(\boldsymbol{\pi}_t^1; \boldsymbol{\lambda}_t)$ is the regret arises from the population difference of stationary optimal policy ($\boldsymbol{\pi}^*$) and the policy of interest ($\boldsymbol{\pi}_t$).
- $\Delta R(\boldsymbol{\pi}_t^1, \boldsymbol{\pi}_t) := R(\boldsymbol{\pi}_t^1; \boldsymbol{\lambda}_t) - R(\boldsymbol{\pi}_t; \boldsymbol{\lambda}_t)$ is the one-step regret of the policy.

- const. $= R(\boldsymbol{\pi}_t^{1,*}; \boldsymbol{\lambda}_t^*) - R(\boldsymbol{\pi}_t^*; \boldsymbol{\lambda}_t^*)$ is the one-step regret of the stationally optimal policy. This term does not depend on $\boldsymbol{\pi}$ and only depends on $\boldsymbol{\pi}^*$.

Then, let $\boldsymbol{\lambda}_t^{\pi}$ to be the population dynamics of a stationary policy $\boldsymbol{\pi}$. From the assumption about the policy convergence,

$$\forall \delta, \delta' > 0, \ \exists T_0 \in \mathbb{Z}, \ \text{s.t.,} \ \forall t > T_0, \ D(\boldsymbol{\pi}, \boldsymbol{\pi}_t) < \delta' \ \text{and} \ D(\boldsymbol{\lambda}_t^{\pi}, \boldsymbol{\lambda}_t) < \delta$$

holds true. Thus, we have $\frac{1}{T} \sum_{t=1}^{T} \Delta R(\boldsymbol{\lambda}_t^{\pi}, \boldsymbol{\lambda}_t) \leq \mathcal{O}(\delta/T)$. Therefore,

$$\begin{aligned}
\text{Regret}(\boldsymbol{\pi}) &= \frac{1}{T} \sum_{t=1}^{T} \left( R(\boldsymbol{\pi}^*; \boldsymbol{\lambda}_t^*) - R(\boldsymbol{\pi}_t; \boldsymbol{\lambda}_t) \right) \\
&= \frac{1}{T} \sum_{t=1}^{T} \left( \Delta R(\boldsymbol{\lambda}_t^*, \boldsymbol{\lambda}_t) + \Delta R(\boldsymbol{\pi}_t^1, \boldsymbol{\pi}_t) + \text{const.} \right) \\
&= \frac{1}{T} \sum_{t=1}^{T} \left( \Delta R(\boldsymbol{\lambda}_t^*, \boldsymbol{\lambda}_t^{\pi}) + \Delta R(\boldsymbol{\lambda}_t^{\pi}, \boldsymbol{\lambda}_t) + \Delta R(\boldsymbol{\pi}_t^1, \boldsymbol{\pi}_t) \right) + \text{const.} \\
&= \underbrace{\frac{1}{T} \sum_{t=1}^{T} \Delta R(\boldsymbol{\lambda}_t^*, \boldsymbol{\lambda}_t^{\pi})}_{(1)} + \underbrace{\frac{1}{T} \sum_{t=1}^{T} \Delta R(\boldsymbol{\pi}_t^1, \boldsymbol{\pi}_t)}_{(2)} + \mathcal{O}\left(\frac{\delta}{T}\right) + \text{const.}
\end{aligned}$$

$\square$

### B.4. Proof of Theorem 3

*Proof.* When $f, \bar{\lambda}$ are linear functions, from Theorem 1 we know that the NE of $\mathcal{G}(\boldsymbol{\pi}, B, f, \bar{\lambda})$ exists and is unique. For any fixed $\boldsymbol{\pi}$, let $\boldsymbol{\lambda}_{\infty} = (\boldsymbol{\lambda}_u^*, \boldsymbol{\lambda}_c^*)$ denote the NE obtained under $\boldsymbol{\pi}$. By Proposition 1, $(\boldsymbol{\lambda}_u^*, \boldsymbol{\lambda}_c^*)$ is also the unique stable fixed point of system (4),(5) and therefore satisfies

$$\lambda_k^{(u)*} = \bar{\lambda}_k \left( \sum_{l=1}^{L} \pi_{l,k} \left( b_{l,k} + f(\lambda_l^{(c)*}) \right) \right), \quad \text{and} \quad \lambda_l^{(c)*} = \bar{\lambda}_l \left( \sum_{k=1}^{K} \pi_{l,k} \lambda_k^{(u)*} \right), \quad \forall 1 \leq l \leq L, 1 \leq k \leq K. \quad (18)$$

Next, we derive the closed-form of $(\boldsymbol{\lambda}_u^*, \boldsymbol{\lambda}_c^*)$. Suppose

$$f(x) = a_0 x + b_0, \quad \bar{\lambda}_k(x) = a_1 x + b_1, \quad \bar{\lambda}_l(x) = a_2 x + b_2, \quad a_0, a_1, a_2 > 0.$$

From Eq. (18) we know $(\boldsymbol{\lambda}_u^*, \boldsymbol{\lambda}_c^*)$ is the unique solution to the following linear system

$$\begin{bmatrix} I_K & -a_0 a_1 \boldsymbol{\pi}^{\top} \\ -a_2 \boldsymbol{\pi} & I_L \end{bmatrix} \begin{bmatrix} \boldsymbol{\lambda}_u^* \\ \boldsymbol{\lambda}_c^* \end{bmatrix} = \left[ a_1 \sum_{l=1}^{L} \pi_{l,1}(b_{l,1} + b_0) + b_1, \cdots, a_1 \sum_{l=1}^{L} \pi_{l,K}(b_{l,K} + b_0) + b_1, b_2, \cdots, b_2 \right]^{\top}, \quad (19)$$

where $I_K, I_L$ denote the identity matrices of sizes $K, L$. Since $\sum_{l=1}^{L} \pi_{l,k} = 1$, we have

$$a_1 \sum_{l=1}^{L} \pi_{l,k}(b_{l,k} + b_0) + b_1 = a_1 \sum_{l=1}^{L} \pi_{l,k} \left( b_{l,k} + b_0 + \frac{b_1}{a_1} \right), \forall 1 \leq k \leq K.$$

Without loss of generality, we let $b_0, b_1 = 0$ hereafter, since we can always absorb the term $b_0 + \frac{b_1}{a_1}$ into $B$ by letting $\tilde{b}_{l,k} = b_{l,k} + b_0 + \frac{b_1}{a_1}$. As a result, from Eq. (19) we can obtain the closed-form solution for $(\boldsymbol{\lambda}_u^*, \boldsymbol{\lambda}_c^*)$ as follows:

$$\begin{aligned}
\begin{bmatrix} \boldsymbol{\lambda}_u^* \\ \boldsymbol{\lambda}_c^* \end{bmatrix} &= \begin{bmatrix} (I_K - a_0 a_1 a_2 \boldsymbol{\pi}^{\top} \boldsymbol{\pi})^{-1} & (I_K - a_0 a_1 a_2 \boldsymbol{\pi}^{\top} \boldsymbol{\pi})^{-1} a_0 a_1 \boldsymbol{\pi}^{\top} \\ a_2 \boldsymbol{\pi} (I_K - a_0 a_1 a_2 \boldsymbol{\pi}^{\top} \boldsymbol{\pi})^{-1} & I_L + a_0 a_1 a_2 \boldsymbol{\pi} (I_K - a_0 a_1 a_2 \boldsymbol{\pi}^{\top} \boldsymbol{\pi})^{-1} \boldsymbol{\pi}^{\top} \end{bmatrix} \\
&\quad \cdot \left[ a_1 \sum_{l=1}^{L} \pi_{l,1} b_{l,1}, \cdots, a_1 \sum_{l=1}^{L} \pi_{l,u} b_{l,u}, b_2, \cdots, b_2 \right]^{\top},
\end{aligned} \quad (20)$$

where $I_K - a_0 a_1 a_2 \boldsymbol{\pi}^\top \boldsymbol{\pi}$ is a positive definite matrix.

On the other hand, the user-side social welfare can be rewritten into

$$
\begin{aligned}
R(\boldsymbol{\pi}; (\boldsymbol{\lambda}_u^*, \boldsymbol{\lambda}_c^*)) &= \sum_{k=1}^K \lambda_k^{(u)*} \sum_{l=1}^L \pi_{l,k}(b_{l,k} + f(\lambda_l^{(c)*})) \\
&= \boldsymbol{\lambda}_u^{*\top} \cdot \left[ \sum_{l=1}^L \pi_{l,1} b_{l,1}, \cdots, \sum_{l=1}^L \pi_{l,K} b_{l,K} \right]^\top + a_0 \boldsymbol{\lambda}_u^{*\top} \boldsymbol{\pi}^\top \boldsymbol{\lambda}_c^* \\
&= \frac{1}{a_1} \boldsymbol{\lambda}_u^{*\top} (\boldsymbol{\lambda}_u^* - a_0 a_1 \boldsymbol{\pi}^\top \boldsymbol{\lambda}_c^*) + a_0 \boldsymbol{\lambda}_u^{*\top} \boldsymbol{\pi}^\top \boldsymbol{\lambda}_c^* && \text{by Eq. (19)} \\
&= \frac{\boldsymbol{\lambda}_u^{*\top} \boldsymbol{\lambda}_u^*}{a_1}.
\end{aligned} \tag{21}
$$

From Eq. (20) we also have

$$
\begin{aligned}
\boldsymbol{\lambda}_u^* &= (I_K - a_0 a_1 a_2 \boldsymbol{\pi}^\top \boldsymbol{\pi})^{-1} \cdot a_1 \cdot \left[ \sum_{l=1}^L \pi_{l,1} b_{l,1}, \cdots, \sum_{l=1}^L \pi_{l,K} b_{l,K} \right]^\top \\
&\quad + (I_K - a_0 a_1 a_2 \boldsymbol{\pi}^\top \boldsymbol{\pi})^{-1} a_0 a_1 \boldsymbol{\pi}^\top \cdot b_2 \mathbf{1}_L \\
&= a_1 (I_K - a_0 a_1 a_2 \boldsymbol{\pi}^\top \boldsymbol{\pi})^{-1} \left[ \mathrm{diag}(\boldsymbol{\pi}^\top B) + a_0 b_2 \mathbf{1}_K \right], && \text{by } \boldsymbol{\pi}^\top \mathbf{1}_L = \mathbf{1}_K.
\end{aligned} \tag{22}
$$

Plug Eq. (22) into Eq. (21), we obtain the following explicit expression of $R$ for any $\boldsymbol{\pi}$ and $B$:

$$
R(\boldsymbol{\pi}; \boldsymbol{\lambda}_\infty) = R(\boldsymbol{\pi}; (\boldsymbol{\lambda}_u^*, \boldsymbol{\lambda}_c^*)) = a_1 \left\| (I_K - a_0 a_1 a_2 \boldsymbol{\pi}^\top \boldsymbol{\pi})^{-1} \left[ \mathrm{diag}(\boldsymbol{\pi}^\top B) + a_0 b_2 \mathbf{1}_K \right] \right\|_2^2, \tag{23}
$$

where $\mathbf{1}_K = (1, \cdots, 1)^\top$ is a column vector of length $K$. Let $\sigma_{\max}(\cdot)$ and $\sigma_{\min}(\cdot)$ denotes the largest and the smallest eigenvalue of a matrix. Then Eq. (23) implies

$$
R_-(\boldsymbol{\pi}; \boldsymbol{\lambda}_\infty) \triangleq \frac{a_1 \left\| \mathrm{diag}(\boldsymbol{\pi}^\top B) + a_0 b_2 \mathbf{1}_K \right\|_2^2}{\sigma_{\max}^2(I_K - a_0 a_1 a_2 \boldsymbol{\pi}^\top \boldsymbol{\pi})} \leq R(\boldsymbol{\pi}; \boldsymbol{\lambda}_\infty) \leq \frac{a_1 \left\| \mathrm{diag}(\boldsymbol{\pi}^\top B) + a_0 b_2 \mathbf{1}_K \right\|_2^2}{\sigma_{\min}^2(I_K - a_0 a_1 a_2 \boldsymbol{\pi}^\top \boldsymbol{\pi})} \triangleq R_+(\boldsymbol{\pi}; \boldsymbol{\lambda}_\infty). \tag{24}
$$

Next, we consider any $\epsilon$-greedy policy $\boldsymbol{\pi}^{(\epsilon)}$ w.r.t. $B$ and show that both $R_-(\boldsymbol{\pi}^{(\epsilon)}; \boldsymbol{\lambda}_\infty^{(\epsilon)})$ and $R_+(\boldsymbol{\pi}^{(\epsilon)}; \boldsymbol{\lambda}_\infty^{(\epsilon)})$ as functions of $\epsilon$ are monotonically decreasing in $\epsilon \in [0, 1]$. Without loss of generality, we may assume the greedy recommendation policy $\boldsymbol{\pi}_0$ has the following form:

$$
\boldsymbol{\pi}_0 = \mathbb{I}\{b_{l,k} = \arg \max_{1 \leq i \leq L} b_{i,k}\}]_{L \times K} = \begin{bmatrix} \mathbf{1}_{K_1} & \mathbf{0} & \cdots & \mathbf{0} & \mathbf{0} \\ \mathbf{0} & \mathbf{1}_{K_2} & \cdots & \mathbf{0} & \mathbf{0} \\ \vdots & \vdots & & \vdots & \vdots \\ \mathbf{0} & \mathbf{0} & \cdots & \mathbf{0} & \mathbf{1}_{K_m} \\ \mathbf{0} & \mathbf{0} & \cdots & \mathbf{0} & \mathbf{0} \end{bmatrix},
$$

i.e., all user groups are clustered into $m$ sub-groups and each has size $K_m$. Each user within a sub-group prefers the same content group and users from different sub-groups prefer different content groups. The total number of user sub-groups $m$ satisfies $1 \leq m \leq L$, $K = K_1 + \cdots + K_m$, and $K_1 \geq \cdots \geq K_m$.

Denote $\boldsymbol{b}_0 = \mathrm{diag}(\boldsymbol{\pi}_0^\top B) = [\max_{1 \leq l \leq L}\{b_{l,k}\}]_{k=1}^k$, and $\boldsymbol{b}_1 = [\frac{1}{L} \sum_{l=1}^L b_{l,k}]_{k=1}^K$. By plugging $\boldsymbol{\pi}_\epsilon = (1 - \epsilon)\boldsymbol{\pi}_0 + \frac{\epsilon}{L} \mathbf{1}_{L \times K}$ into Eq. (23), we obtain

$$
\mathrm{diag}(\boldsymbol{\pi}_\epsilon^\top B) + a_0 b_2 \mathbf{1}_K = (1 - \epsilon)\boldsymbol{b}_0 + \epsilon \boldsymbol{b}_1 + a_0 b_2 \mathbf{1}_K.
$$

Since $\boldsymbol{b}_0 \geq \boldsymbol{b}_1$ elementary-wise, we conclude that $\|\mathrm{diag}(\boldsymbol{\pi}_\epsilon^\top B) + a_0 b_2 \mathbf{1}_K\|_2$ as a function of $\epsilon$ is decreasing in $[0, 1]$.

On the other hand, direct calculation shows

$$I_K - a_0 a_1 a_2 \boldsymbol{\pi}_\epsilon^\top \boldsymbol{\pi}_\epsilon = I_k - a_0 a_1 a_2 \left[ (1-\epsilon)\boldsymbol{\pi}_0^\top + \frac{\epsilon}{L}\mathbf{1}_{K \times L} \right] \cdot \left[ (1-\epsilon)\boldsymbol{\pi}_0 + \frac{\epsilon}{L}\mathbf{1}_{L \times K} \right]$$

$$= I_K - a_0 a_1 a_2 (1-\epsilon)^2 \boldsymbol{\pi}_0^\top \boldsymbol{\pi}_0 - \frac{a_0 a_1 a_2 \epsilon(2-\epsilon)}{L}\mathbf{1}_{K \times K}.$$

Given the explicit form of the block matrix $\pi_0$, we can directly compute the smallest and the largest eigenvalues of matrix $I_K - a_0 a_1 a_2 (1-\epsilon)^2 \boldsymbol{\pi}_0^\top \boldsymbol{\pi}_0$ as followings:

$$\sigma_{\max}(I_K - a_0 a_1 a_2 (1-\epsilon)^2 \boldsymbol{\pi}_0^\top \boldsymbol{\pi}_0) = 1,$$
$$\sigma_{\min}(I_K - a_0 a_1 a_2 (1-\epsilon)^2 \boldsymbol{\pi}_0^\top \boldsymbol{\pi}_0) = 1 - a_0 a_1 a_2 (1-\epsilon)^2 K_1.$$

In addition, from Weyl's inequality (Fan, 1949; Bunch et al., 1978), we conclude that

$$\sigma_{\max}(I_K - a_0 a_1 a_2 \boldsymbol{\pi}_\epsilon^\top \boldsymbol{\pi}_\epsilon) = 1,$$
$$\sigma_{\min}(I_K - a_0 a_1 a_2 \boldsymbol{\pi}_\epsilon^\top \boldsymbol{\pi}_\epsilon) \geq \sigma_{\min}(I_K - a_0 a_1 a_2 (1-\epsilon)^2 \boldsymbol{\pi}_0^\top \boldsymbol{\pi}_0) - \sigma_{\max}\left( \frac{a_0 a_1 a_2 \epsilon(2-\epsilon)}{L}\mathbf{1}_{K \times K} \right)$$
$$= 1 - a_0 a_1 a_2 (1-\epsilon)^2 K_1 - \frac{a_0 a_1 a_2 \epsilon(2-\epsilon)K}{L}$$
$$= 1 - a_0 a_1 a_2 K_1 + a_0 a_1 a_2 \epsilon(2-\epsilon)\left( K_1 - \frac{K}{L} \right). \tag{25}$$

Note that by the definition of $K_1$, it holds that $K_1 \geq \frac{K}{m} \geq \frac{K}{L}$. Hence, the lower bound of $\sigma_{\min}$ in Eq. (25) is an increasing function in $\epsilon \in [0, 1]$.

Hence, we have

$$R_-(\boldsymbol{\pi}^{(\epsilon)}; \boldsymbol{\lambda}_\infty^{(\epsilon)}) = a_1 \|(1-\epsilon)\boldsymbol{b}_0 + \epsilon \boldsymbol{b}_1 + a_0 b_2 \mathbf{1}_K\|_2^2, \tag{26}$$

$$R_+(\boldsymbol{\pi}^{(\epsilon)}; \boldsymbol{\lambda}_\infty^{(\epsilon)}) \leq \frac{a_1 \|(1-\epsilon)\boldsymbol{b}_0 + \epsilon \boldsymbol{b}_1 + a_0 b_2 \mathbf{1}_K\|_2^2}{\left( 1 - a_0 a_1 a_2 K_1 - a_0 a_1 a_2 \epsilon(2-\epsilon)\left( K_1 - \frac{K}{L} \right) \right)^2}, \tag{27}$$

and the RHS of both Eqs. (26) and (27) are decreasing functions of $\epsilon$ in $[0, 1]$.

Take $g(\epsilon) = a_1\|(1-\epsilon)\boldsymbol{b}_0 + \epsilon \boldsymbol{b}_1 + a_0 b_2 \mathbf{1}_K\|_2^2$, and $h(\epsilon) = \left( 1 - a_0 a_1 a_2 K_1 - a_0 a_1 a_2 \epsilon(2-\epsilon)\left( K_1 - \frac{K}{L} \right) \right)^{-2}$, we conclude that

$$g(\epsilon) \leq R(\boldsymbol{\pi}^{(\epsilon)}; \boldsymbol{\lambda}_\infty^{(\epsilon)}) \leq g(\epsilon)h(\epsilon),$$

and $h(\epsilon) = \left( 1 - a_0 a_1 a_2 K_1 - a_0 a_1 a_2 \epsilon(2-\epsilon)\left( K_1 - \frac{K}{L} \right) \right)^{-2} < (1 - 2a_0 a_1 a_2 K)^{-2}$. Since by definition $(\nabla_{\lambda_l} f_{k,l})(\nabla_{e_l} \bar{\lambda}_l)(\nabla_{s_k} \bar{\lambda}_k) = a_0 a_1 a_2$, our claim holds. $\square$

### B.5. Proof of Proposition 2

*Proof.* Consider a two-sided system with $K = 1, L = 2, B = [1, 0.9]^\top$ and

$$\bar{\lambda}_1^u(x) = a_0 x, \bar{\lambda}_1^c(x) = a_1 x, \bar{\lambda}_2^c(x) = a_2 x, f_1(x) = b_1 x, f_2(x) = b_2 x.$$

According to Theorem 1, the NE of the system exists and is unique when $a_0, a_1, a_2 > 0$ are sufficiently small. Moreover, $(\lambda_1^{(u)}, \lambda_1^{(c)}, \lambda_2^{(c)})$ at the NE satisfies

$$\lambda_1^{(u)} = \bar{\lambda}_1^{(u)}(\pi_{11} f_1(\lambda_1^{(c)}) + \pi_{21} f_2(\lambda_2^{(c)}) + \pi_{11} + 0.9\pi_{21}),$$
$$\lambda_1^{(c)} = \bar{\lambda}_1^{(c)}(\pi_{11}\lambda_1^{(u)}),$$
$$\lambda_2^{(c)} = \bar{\lambda}_2^{(c)}(\pi_{21}\lambda_1^{(u)}),$$

which is equivalent to

$$\begin{cases} \lambda_1^{(u)} = a_0(b_1\pi_{11}\lambda_1^{(c)} + b_2\pi_{21}\lambda_2^{(c)} + \pi_{11} + 0.9\pi_{21}), \\ \lambda_1^{(c)} = a_1\pi_{11}\lambda_1^{(u)}, \\ \lambda_2^{(c)} = a_2\pi_{21}\lambda_1^{(u)}. \end{cases} \tag{28}$$

Plugin the last two equations into the first one in Eq. (28), we obtain that

$$\lambda_1^{(u)} = a_0(a_1b_1\pi_{11}^2\lambda_1^{(u)} + a_2b_2\pi_{21}^2\lambda_1^{(u)} + \pi_{11} + 0.9\pi_{21}),$$

and therefore

$$\begin{aligned} \lambda_1^{(u)} &= \frac{a_0(\pi_{11} + 0.9\pi_{21})}{1 - a_0a_1b_1\pi_{11}^2 - a_0a_2b_2\pi_{21}^2} \\ &= \frac{a_0(0.9 + 0.1\pi_{11})}{1 - a_0a_1b_1\pi_{11}^2 - a_0a_2b_2(1 - \pi_{11})^2}. \end{aligned}$$

Now we can write $R(\boldsymbol{\pi}; \boldsymbol{\lambda}^*)$ as a function of $\pi_{11}$ as the following:

$$\begin{aligned} R(\boldsymbol{\pi}; \boldsymbol{\lambda}^*) &= \lambda_1^{(u)}(\pi_{11}f_1(\lambda_1^{(c)}) + \pi_{21}f_2(\lambda_2^{(c)}) + \pi_{11} + 0.9\pi_{21}) = \frac{1}{a_0}(\lambda_1^{(u)})^2 \\ &= \frac{a_0(0.9 + 0.1\pi_{11})^2}{[1 - a_0a_1b_1\pi_{11}^2 - a_0a_2b_2(1 - \pi_{11})^2]^2}. \end{aligned}$$

Take $b_1 = 0, a_0a_2b_2 = 0.4$, we have

$$\sqrt{\frac{R(\boldsymbol{\pi}, \boldsymbol{\lambda}^*)}{a_0}} = \frac{0.9 + 0.1\pi_{11}}{1 - 0.4(1 - \pi_{11})^2} \triangleq \tilde{R}(\pi_{11}), \tag{29}$$

and it is easy to verify that the RHS of Eq. (29) is not achieved at $\pi_{11} = 1$. In fact, for any $\pi'_{11} < 0.7$, it holds that $\tilde{R}(\pi'_{11}) > \tilde{R}(1) = 1$. This means the greedy policy $[\pi_{11}, \pi_{21}] = [1, 0]$ is not optimal in this example.

$\square$

## C. Additional experiment settings and observations

Here, we report additional details of the experiment settings and results.

**Difference of population effects in the synthetic and real-world experiments.** Figure 6 shows the population effect used in the synthetic experiment, defined by Eq. (13). The biggest difference between the synthetic and real-world experiment setting is that we observe saturation of the population effects as an early stage of the provider population growth, i.e., around $\lambda_l = 100$, which is also a reasonable phenomenon in real-world situations. Therefore, in the synthetic experiment, it is important to distribute the content exposure among multiple subgroups to receive high population effects in many different provider groups. Thus, even the uniform policy outperforms the myopic-policy in this setting.[6] In contrast, when using KuaiRec dataset (Gao et al., 2022), the situation is milder than the synthetic experiment, and therefore the myopic policy works well in the real-world experiment. Together, our synthetic and real-world experiments show that the proposed look-ahead policy performs reasonably well in two different configurations. This is because seeking for both (immediate) viewer utility ($s_k$) and provider exposure ($e_l$) is important to maximize the look-ahead objective, which depends on reference populations $\bar{\lambda}(s_k)$ and $\bar{\lambda}(e_l)$.

---

[6]Throughout the experiments, we found that there are tradeoffs in concentrating and distributing exposures, and which is better often depends on the problem instance. For example, suppose the special case where viewers do not change their population ($\eta_k = 0, \forall k \in [K]$) and the total viewer population is fixed to 100. Then, consider a scenario with 100 provider groups. In this situation, distributing exposure among different subgroups can result in expected exposure of 1 for each provider group. In such cases, concentrating the exposure to one provider group can be a better strategy than distributing allocation for the total population growth. This is why myopic policy performs well in some scenarios, and our argument is that the proposed method can work adaptively well (i.e., at least better or competitive than both myopic and uniform) regardless if the myopic policy succeeds or falls short.

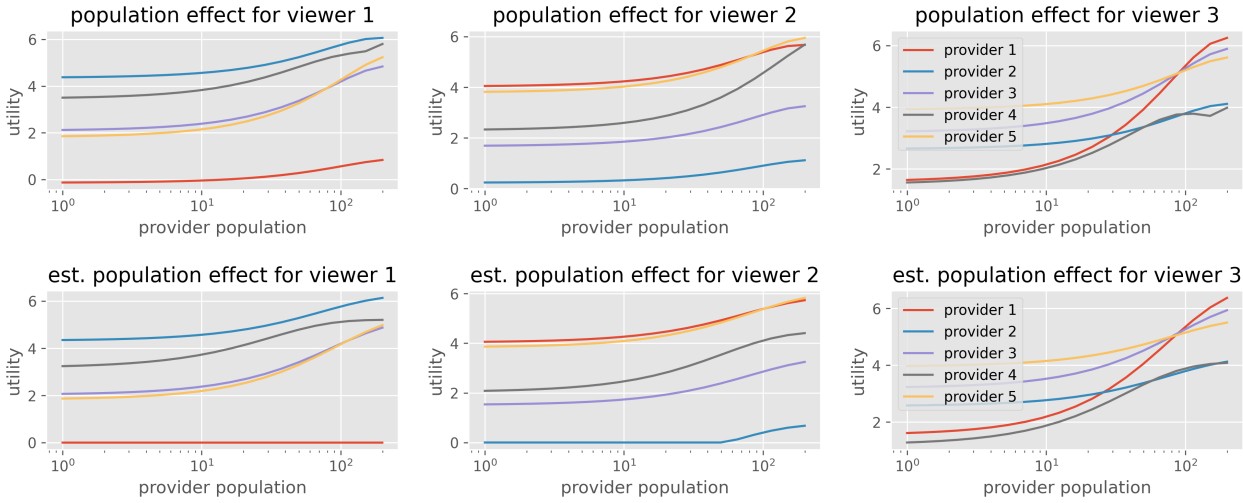

*Figure 6.* **Visualization of the population effects in the synthetic experiment.** We randomly sample the scaler and temperature parameter of the sigmoid function from a normal distribution for each content-quality feature pair as described in Section 6. The resulting quality vector is provider-dependent, and the population effects are heterogeneous across viewer-provider pairs.

*Figure 7.* **Comparing the true and estimated population effect in the real-world experiment.** (Top) True population effect used in the real-world experiment (the same figure as Figure 4 in the main text). (Bottom) Population effect learned by the long-term optimal policy at the final timestep.

**Estimation results of the dynamics and population effect functions.** We also report how the dynamics estimation works in the real-world experiment in Figures 7 and 8. While we initialize the population effect and dynamics estimation with a homogeneous function across viewer-provider pairs, the results demonstrate that our estimation scheme provides an accurate estimation of heterogeneous functions by using the dynamics logs in the rollout process. We also observe that the policy optimization results in the main text (w/ population effect and dynamics estimation) are quite similar to those without dynamics estimation (i.e., using the true dynamics) in the experiment.

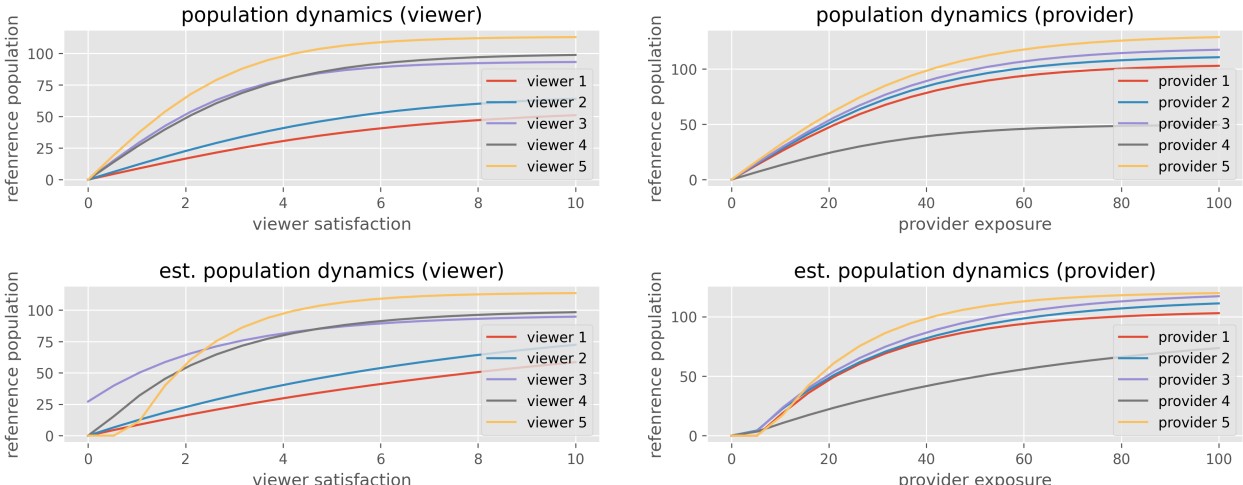

*Figure 8.* **Comparing the true and estimated population dynamics in the real-world experiment.** (Top) True population dynamics simulated in the real-world experiment. (Bottom) Population dynamics learned by the long-term optimal policy at the final timestep.

