# OpenReview forum: "Policy Design for Two-sided Platforms with Participation Dynamics"
_ICML.cc/2025/Conference — ICML 2025 poster_

### Official Review · Reviewer_g8MP · 2025-02-28

**Overall Recommendation:** 2

**Summary:**

This paper studies the effect of matching policies in two-sided platforms, taking the evolution of both viewer and provider sides into consideration.
The authors show that myopic matching policies are only optimal in strong assumptions and appeared to be suboptimal at other cases.
The authors propose a new matching policy, called "look-ahead", that appears to perform better and more stable than myopic and uniform policies in experiments.

**Claims And Evidence:**

Yes.

**Essential References Not Discussed:**

N/A

**Experimental Designs Or Analyses:**

Yes. The experiment results support the main findings of this paper (myopic matching policies are suboptimal).

**Methods And Evaluation Criteria:**

Yes.

**Other Comments Or Suggestions:**

C1: In line 194, it seems that "Theorem 4" should be "Theorem 1".
C2: In Equation (3), it seems that the introduction of $b_{k,l}$ is actually unnecessary because it can be incorporated into $f_{k,l}(\lambda_l)$.

**Other Strengths And Weaknesses:**

* Strengths:

S1: The presentation is good and easy to follow.
S2: The dynamic provided in Equation (4), (5), and the game-theoretic interpretation in Equation (6), are novel and interesting.

* Weaknesses:

W1: Many results stated in this paper are straightforward and not surprising. For examples,

1) Proposition 1 is a straightforward statement of fact.
2) In Theorem 1, the existence and non-uniqueness of NE are well-established throughout game theory and not surprising. The second statement of Theorem 1 is standard in optimization theory.
3) The stability in Proposition 2 is also straightforward (same intuitions with those in Theorem 1).
4) The derivation of Theorem 2 seems to be straight-forward. The theorem actually states some intuition of "decomposition".
5) In Theorem 3, the assumption in first statement is strong (K=1). For second statement of Theorem 3, the result can say nothing about whether $R$ is monotone decreasing over $\varepsilon$, even if $h(\varepsilon)$ can be arbitrarily close to 1. In addition, Theorem 3 only considers the interpolation between greedy policy and uniform policy, and ignores the high-dimensional policy space.
6) In Proposition 3, the suboptimality of myopic policy is standard in almost every field.

W2: In RHS of line 176, there is no definition about the concept used in "$\lambda_{eq}$ is stable".
W3: In the paragraph below Proposition 2, the explanation about the polarized equilibrium of exposure-concentrated policy is hard-to-believe, with no theoretical and empirical evidence.
W4: The proposed policy, "look-ahead policy", is straight-forward to design. Besides, no theoretical guarantees are provided for this policy.
W5: In experiments, the only considered baselines are uniform and myopic policies, both of which are weak.

In other hands, the only contribution the authors claim seems to be that this paper is the first to consider two-side dynamics in this literature, and I believe it indeed is. However, the results derived in this paper (e.g., non-uniqueness and stability of Nash Equilibriums, suboptimality of myopic policy) are likely to hold and be discovered in many other settings. So it seems that the only contribution is to extend these straight-forward results to a new setting, and such contribution I think does not warrant an ICML acceptance.

**Questions For Authors:**

I have no questions.

**Relation To Broader Scientific Literature:**

The author mentioned that this paper is the first paper that studies matching policies in two-side platforms, taking the evolution of both viewer and provider sides into consideration.
Unfortunately, I'm unfamiliar with the literature and thus unqualified to evaluate this statement.

**Theoretical Claims:**

I find some steps confusing in the proof of Proposition 2.

* In line 645-646, what do you mean by saying $\nabla _\lambda S$ and why can you say $\lambda _{t+1} - \lambda _{eq} = (\nabla _\lambda S) (\lambda _t - \lambda _{eq})$? I can only derive $\lambda _{t+1} - \lambda _{eq} = S(\lambda _t ) - S(\lambda _{eq})$ from the information given up to there. I also note that $S(\lambda)$ is a vector instead of a real number, so one can not directly use the statement $S(\lambda _t ) - S(\lambda _{eq}) = (\nabla _\lambda S(\lambda^*)) (\lambda _t - \lambda _{eq})$ for some $\lambda^*$.

* In line 654-658, I'm confused about the derivation, it seems that $\{A _{1,1} \} _{k,k'} = -1 \mathbb{I} \{k = k'\}$, same for $\{A _{2,2}\}$. For $\{A _{1,2}\} _{k,l}$, it should be $\{A _{1,2}\} _{k,l} = \nabla _{\lambda _l} \bar{\lambda _k}(\sum _{l'} \pi _{k, l'} (b _{k,l'} + f _{k,l'}(\lambda _{l'})))$, and no $\eta _k$ ($\eta _l$) term should appear in this term.

---

> ### Author Rebuttal · Authors · 2025-04-01
>
> We would like to thank the reviewer for their time and efforts on the review. We respond to the key comments and questions below.
>
> ---
>
>
> **(factual discussion)**
>
> ---
>
> > **W3 [polarized equilibrium]**
>
> We clarify the reasoning behind the paragraph below Proposition 2 step by step. First, consider the case where $f$ and $\bar{\lambda}$ are monotonically increasing concave functions. This means that the upper bound of the first order derivative, $C_1$ and $C_2$ are monotonically decreasing to the viewer/provider populations.
>
> Then, let us assess if a polarized equilibrium can be allowed (i.e., satisfies Ineq. (8)) given some specific policies. When the equilibrium population is polarized, this means that some population diminishes, and thus $C_1$ and $C_2$ can be large. Therefore, RHS of Ineq. (8) becomes small (e.g., near-zero). While the exposure-fair policy may exclude this point due to the violation of Ineq. (8), the exposure-concentrated policy can allow this point (Ineq. (8) is satisfied) by letting the LHS of Ineq. (8) to be (near-)zero or smaller than RHS.
>
> ---
>
>
> > **Theoretical guarantees of look-ahead policy**.
>
> Please also refer to the rebuttals for Reviewer E67u.
>
> ---
>
> > **Baselines in experiments**.
>
> Please refer to the rebuttals for Reviewer 54YD.
>
> ---
>
> **(contributions)**
>
> ---
>
> > **W1 [why our results are not trivial]**.
> While our results may appear straightforward, we respectfully disagree that they are *trivial* and would like to clarify why they are novel.
>
> **Theorem 1**, a notable insight of our result is the guaranteed existence of pure NE in a two-sided market setting. This contrasts with findings in relevant prior works, such as [1], [2], and [3], which study one-sided markets involving only strategic content creators and demonstrate that pure NEs may fail to exist or only exist under restrictive conditions. Our result highlights a meaningful and somewhat surprising message: when the passive resource (e.g., user attention) in previous models becomes an active participant (as in our two-sided market), the game structure always ensures the existence of pure NE. We recognize this as an important conceptual contribution and will highlight it more clearly in the revised version.
>
>
> **Proposition 2**, we would like to emphasize two non-trivial messages. First, it is well-established (e.g., [4], [5]) that the existence of pure NE does NOT automatically imply convergence under multi-agent gradient dynamics; in fact, such dynamics may get stuck in local NEs [5] or even non-Nash stationary points [4]. In contrast, our model establishes that gradient dynamics provably converge under certain conditions, which is an important theoretical contribution. Second, the sufficient conditions we identify for convergence—namely, weak population effects and more equitable exposure allocation—are not only intuitive but also offer actionable insights for policy design. We briefly discussed these points in the paragraph following Eq. (8) and will make them more explicit in the revision.
>
>
> **Theorem 3**, we agree that fully characterizing the optimality of myopic-greedy policies is technically challenging. However, our synthetic experiments in Section 6 do well support our theoretical insight in Section 4. In fact, the main purpose of Section 4 is to conceptually illustrate the scenarios in which myopic-greedy policies perform well versus when they fall short. As a result, Section 4 serves as a motivating example that leads naturally into our exploration of optimal policy design in subsequent sections.
>
>
> [1] Modeling content creator incentives on algorithm-curated platforms.
>
> [2] Supplyside equilibria in recommender systems.
>
> [3] How Bad is Top-$ K $ Recommendation under Competing Content Creators?
>
> [4] On finding local nash equilibria (and only local nash equilibria) in zero-sum games.
>
> [5] User welfare optimization in recommender systems with competing content creators.
>
> ---
>
> > **Additional clarifications about our contributions**.
>
> We would like to kindly argue that these contributions are actually acknowledged by reviewers KANq and 54YD as “this paper studied an interesting problem ... The theoretical and numerical results both appear sound and provide interesting insights.” “the model is interesting, as are the results.”
>
> If the proposed method appears to be straightforward, it means that our method is well-motivated by the theoretical analysis provided in advance in the paper (especially Theorem 2). Moreover, as we have discussed about Theorem 1 and Proposition 2, non-uniqueness and stability of NEs do NOT actually hold in many other similar settings. Providing theoretical evidence for a seemingly reasonable hypothesis, highlighting the overlooked problem, and giving a reasonable explanation is crucial for scientific research. We would appreciate it if the reviewer could acknowledge our contribution regarding this point.

---

> > ### Comment · Reviewer_g8MP · 2025-04-04
> >
> > Thank you for your response. I apologize for my inappropriate words in my initial comments, and I have changed my words.
> >
> > I would like to give further comments. Regarding the results about the existence and non-uniqueness of pure NEs, since I'm not familiar with this field, I do not know why these results can not be established in other settings. But these results are well-established in concave games [1]. The game considered in this submission is exactly a (strongly) concave game. For this reason, the results about the existence and non-uniqueness of pure NEs seem to be straightforward from my point of view.
> >
> > [1] Rosen, J. Ben. "Existence and uniqueness of equilibrium points for concave n-person games." Econometrica: Journal of the Econometric Society (1965): 520-534.
> >
> > In addition, I found some proof details confusing, and I have updated my initial review.
> >
> > Finally, I believe that the problem studied in this paper is important and novel. But my main concerns lie in that the results seem to be straight-forward and not strong (from my general knowledge of game theory). Furthermore, I believe this concern should not be resolved only because other reviewers think the results are strong. Above all, I will maintain my initial score.

---

### Official Review · Reviewer_54YD · 2025-03-13

**Overall Recommendation:** 4

**Summary:**

In this paper, the authors formulated and studied the dynamics of population effects on two-sided platforms, where viewer and provider populations evolve based on certain rules. Theoretically, the authors show that the myopic-greedy policy can fail to perform well when the population effects are heterogeneous across providers. They further investigated the shortcomings of the myopic policy by decomposing the overall welfare regret into both the policy regret and the population regret, where the latter captures the long-term welfare loss that is not captured by the myopic policy. In response to this, the authors proposed an algorithm that balances policy and population regrets, and show the effectiveness of this approach via synthetic and real-world numerical experiments.

**Claims And Evidence:**

Overall, I think the claims made in this paper are quite convincing. I think the key assumptions used in the model quite realistic and could contribute to understanding how to best balance short-term and long-term welfare in two-sided platforms. The authors also justified their results using both solid theoretical and numerical evidence.

**Essential References Not Discussed:**

N/A

**Experimental Designs Or Analyses:**

- Overall, the experiment results appear sound. I also appreciate that the authors conduct both synthetic and real-world experiments to validate the effectiveness of their approaches.
- I have some questions related to the practicality of the algorithm, which I detailed below.

**Methods And Evaluation Criteria:**

I think it's good that the authors evaluate their method on both synthetic and real-world data. However, I do think they can consider more baselines methods in addition to the myopic and uniform random policy. For example, could some related works that model the population departure be included as baselines?

**Other Comments Or Suggestions:**

N/A

**Other Strengths And Weaknesses:**

Strength:
- As I mentioned above, I think this paper studied an interesting problem that formulate the population dynamics in two-sided platforms. The theoretical and numerical results both appear sound and provide interesting insights.

Weakness:
- It'd be good if the authors can provide more details for the look-ahead policy in Section 5. For example,
    (1) The authors proposed to use the softmax policy as the approximation of $\pi_t^1$; would this impact the performance of the policy?
    (2) It appears that one needs to know the viewer satisfaction between each viewer/provider pair in order to solve the optimization problem for the look-ahead policy. What if the platform lacks such knowledge in a realistic setting?
- I also think the authors should offer more discussions how to best select $\beta$ in real-world applications and whether any theoretical results can be obtained. While the authors claimed that setting $\beta = 1.0$ already appears to work well and thus minimizes the effort to tune this parameter, it's somewhat unrealistic to adopt a fully look-ahead policy in real-world applications, especially when platforms might want to lay more emphasis on short-term outcomes. I wonder if the authors can provide more comments on this aspect. Could you establish any guarantees on short-term outcomes under the fully look-ahead policy? Or, if a platform would like to understand how to select the best $\beta$, is there a reasonable way to do so? I suspect that simple A/B tests won't be helpful because they only capture short-term impacts.

**Questions For Authors:**

See my comments above.

**Relation To Broader Scientific Literature:**

- I believe this work contributes to the broad literature on two-sided platform. It touches on important subjects including balancing the interests of strategic agents and balancing the tradeoff between short-run and long-run welfare. In particular, the topic of how to best evaluate and achieve long-term fairness has not been studied a lot.
- That being said, I think the paper should have a more in-depth discussion on the related works which appear to be missing from the main body as of now.

**Theoretical Claims:**

The theoretical results appear sound, though I did not check the proofs in the appendix.

---

> ### Author Rebuttal · Authors · 2025-04-01
>
> We would like to thank the reviewer for valuable feedback and the acknowledgment of the contributions. We respond to the key comments and questions below.
>
> ---
>
> > **Could some related works that model the population departure be included as baselines?**
>
> Thank you for the great point. Unfortunately, the related works, which model the population departure, cannot be directly applied to our setting. There are two reasons: (1) The existing papers assume that the system dynamics are linear, but we consider the non-linear dynamics. Therefore, their linear programming approach is not solvable in our setting. (2) The existing papers do not model the population increase, and even if the dynamics are linear, it is not trivial how to apply the existing work taking the population increase into account (because we do not have a binary indicator of the population). Our paper proposes the first reasonable baseline for optimizing long-term outcomes under flexible and non-linear dynamics of the population.
>
> ---
>
> > That being said, I think the paper should have a **more in-depth discussion on the related works** which appear to be missing from the main body as of now.
>
> Thank you for the thoughtful suggestion, and we appreciate your point. We will have an additional page in the camera-ready version, and we plan to move the discussion of related work that is currently in the Appendix to the main text upon publication.
>
> ---
>
> > **The authors proposed to use the softmax policy as the approximation of $\pi_t^1$; would this impact the performance of the policy?**
>
> This is a great point. By using the softmax policy as the approximation of $\pi_t^1$, we expect some underestimation of the objective function ($R(\cdot)$ in Eq. (10)). However, as long as the maximizer of this objective function ($\pi$) is the same (which is often the case), the performance of the policy should not change by a large amount.
>
> ---
>
> > **It appears that one needs to know the viewer satisfaction between each viewer/provider pair in order to solve the optimization problem for the look-ahead policy. What if the platform lacks such knowledge in a realistic setting?**
>
> This is also a great question. We consider the scenario where we work on sub-group level, and thus the (expected) satisfaction is usually accessible as a subgroup level, and this is also a standard assumption in existing work [1]. However, when this information is unavailable, we need some additional estimation process of viewer satisfaction using past interaction data. We may then need additional consideration of exploration-exploitation tradeoffs, which should be an interesting future direction.
>
> [1] Optimizing Long-term Social Welfare in Recommender Systems: A Constrained Matching Approach.
>
> ---
>
> > I also think the authors should offer more discussions **how to best select $\beta$ in real-world applications and whether any theoretical results can be obtained**.
>
> > **Could you establish any guarantees on short-term outcomes under the fully look-ahead policy?**
>
> These are indeed important points. While we cannot provide guarantee on the short-term outcomes, it is possible to validate the short-term outcome before deploying the look-ahead policy. Throughout the paper, we consider the situation where the immediate outcome is accessible for both baseline and the proposed methods (following existing work). Thus, once we identify $\pi_t^{(d)}$ using Eq. (10), we can compare the short-term (i.e., myopic) outcome of the proposed method and myopic policies. If the platform aims to guarantee some certain short-term outcome, the platform can adjust the value of $\beta$ to satisfy such constraints.

---

### Official Review · Reviewer_KNAq · 2025-03-14

**Overall Recommendation:** 4

**Summary:**

The paper models the dynamics of a two-sided platform with viewers and providers. Viewers receive satisfaction from watching content assigned to them from providers that they like, and providers receive exposure from having their content assigned to viewers. The population of both the viewers and providers are dynamic, and a higher satisfaction for one group of viewers leads to more viewers of that group joining the platform and a higher exposure to one group of providers leads to more providers from that group joining the platform. The latter also improves the quality of the content of that platform.

They argue that myopic policies that only seek to optimize welfare according to the current population can fail to be optimal long-term. They provide an alternative "look-ahead" policy to do this that is solvable (approximately) via gradient ascent. They show via experiments that their policy performs as good (or better) than the myopic policy in certain scenarios.

**Claims And Evidence:**

Their primary claim is that their look-ahead policy performs better than the myopic policy, hence justifying their model and results. This is convincing and believable. Their first set of experiments appear promising, but their second experiment shows that the myopic policy works just as well, if not slightly better. I am not sure why they haven't discussed more about why this is the case.

**Essential References Not Discussed:**

N/A

**Experimental Designs Or Analyses:**

The experiments appear to be make sense within their framework.

**Methods And Evaluation Criteria:**

Yes.

**Other Comments Or Suggestions:**

It would be good to have some explanation as to why the greedy policy worked so well in the real-world experiments. In fact, the greedy policy seems to have the highest (or almost the highest) welfare throughout all the timesteps.

**Other Strengths And Weaknesses:**

The paper is well-written and the model is interesting, as are the results. I am a bit wary about some of their claims, though. For example, they state, after Theorem 3 and Proposition 3,

>"these results demonstrate that the myopic-greedy policy is optimal only under highly restrictive conditions and emphasize the need for practical solutions accounting for the long-term effect.".

Their results in fact, do not show this. I agree that the situation in Theorem 3 is restrictive. But that is only one scenario where it is optimal. The real-world experiments in fact also show a scenario where the myopic policy is near-optimal. Proposition 3 also says nothing about whether greedy is optimal only in such settings. I do however, believe that a stronger statement can be shown that would suggest something more substantial.

**Questions For Authors:**

N/A

**Relation To Broader Scientific Literature:**

The paper mentions several related works such as Mladenov et al (2020), Huttenlocher et al (2023), Hashimoto et al (2018) that consider dynamics under population change of viewers or providers, but not at the same time.

**Theoretical Claims:**

I only briefly skimmed some of the proofs in the appendix.

---

> ### Author Rebuttal · Authors · 2025-04-01
>
> We would like to thank the reviewer for valuable feedback and the acknowledgment of the contributions. We respond to the key comments and questions below.
>
> ---
>
> > **I am a bit wary about some of their claims**, though. For example, they state, after Theorem 3 and Proposition 3, (overstatement)
>
> Thank you for the thoughtful feedback. We appreciate your point, and we plan to rephrase the pointed-out sentence in the revision as follows.
>
> - (Current) “.. the myopic-greedy policy **is optimal** only under highly restrictive ..".
> - (Revision) “.. the myopic-greedy policy **is guaranteed optimal** only under highly restrictive ..”
>
> ---
>
> > It would be good to have some explanation as to why the greedy policy worked so well in the real-world experiments. In fact, the greedy policy seems to have the highest (or almost the highest) welfare throughout all the timesteps.
>
> Thank you for the great point. Throughout the experiments (including the ones not presented in the paper), we found that there are tradeoffs in concentrating and distributing exposures. Which is better often depends on the problem instance.
>
> For example, suppose the special case where viewers do not change their population ($\eta_k = 0, \forall k \in [K]$) and the total viewer population is fixed to 100. Then, consider a scenario with 100 provider groups. In this situation, distributing exposure among different subgroups can result in expected exposure of 1 for each provider group. In such cases, concentrating the exposure to one provider group can be a better strategy than distributing allocation for the total population growth. This is why myopic policy performs well in some scenarios, and our argument is that the proposed method can work adaptively well (i.e., at least better or competitive than both myopic and uniform) regardless if the myopic policy succeeds or falls short.
>
> Note that the reason for having a small gap between the myopic policy and the proposal in the real-data experiment is that our method involves an optimization process (e.g., estimating dynamics and optimizing policy) and thus has small modeling errors. We hope this answer could resolve your question.

---

### Official Review · Reviewer_E67u · 2025-03-14

**Overall Recommendation:** 1

**Summary:**

This paper proposes and analyzes a theoretical model of population effects of the (e.g. recommendation) policies for two-sided platforms, where there are consumers and producers. The basic intuition is that a recommendation system influences both the short-term satisfaction of the content consumer but also the success of the producers, which may influence the size of the producer pool, creating long-term feedback loops. The model consists of finite sets of consumer and producer types, for which consumers derive utility heterogeneously across types of producers. They compare a “myopic” policy that only optimizes for consumer satisfaction with a “look-ahead” policy that also estimates feedback loops.

**Claims And Evidence:**

The claim that the myopic policy can be sub-optimal is a mild one, since all that is necessary is for there to exist some problem instance where it performs poorly.

There is no theoretical evidence in support of the look-ahead policy. Instead, there are simulations showing that it performs well.

My main criticisms are two-fold: (1) the main conclusion, that platforms should attend to producer welfare as well as consumer welfare, is reasonably intuitive and the paper fails to explore more implications of their model beyond this basic point. (2) there is a lack of theoretical results in support of their proposed look-ahead policy. Given the intuitive and reasonably simple model, I might have expected this kind of result to be tractable.

**Essential References Not Discussed:**

N/A

**Experimental Designs Or Analyses:**

N/A

**Methods And Evaluation Criteria:**

The lack of theoretical results in support of their main conclusion is a weakness.

**Other Comments Or Suggestions:**

N/A

**Other Strengths And Weaknesses:**

N/A

**Questions For Authors:**

What additional insight do we get from multiple simulation experiments? I would have preferred to just have one and then have expanded theoretical analysis. (Or an actual implementation of the policies in a lab/crowdsourced setting to see how they each do.)

**Relation To Broader Scientific Literature:**

The focus on population effects in platforms is interesting, well-motivated and deserving of more study. Their model is similar to others in the prior literature.

**Theoretical Claims:**

N/A

---

> ### Author Rebuttal · Authors · 2025-04-01
>
> We would like to thank the reviewer for their time and effort on the review. We respond to the key comments and questions below.
>
> ---
>
> **(factual discussion)**
>
> ---
>
> > **there is a lack of theoretical results in support of their proposed look-ahead policy**
>
> Thank you for the valuable feedback. We would like to kindly argue that providing a theoretical guarantee to the look-ahead policy is challenging due to two reasons: (1) interactive dynamics between viewers and providers, and (2) non-linearity and non-convexity of the systems.
>
> First, one of the related works, [1] considers the departure of viewers and providers in two-sided platforms. While [1] considers a more restrictive setting than ours in modeling linear dynamics and not modeling the increase in viewer and provider participation, identifying the optimal policy is proven to be an NP-hard problem. Second, guaranteeing sub-optimality in non-linear systems is in general considered a hard problem, and without assuming convexity, we often cannot get a theoretical guarantee [2].
>
> The most related theoretical analysis to our method is [3]. This paper studies minimizing loss function for predictions when the population gradually reacts to the model in a single-sided platform. This paper provides some theoretical guarantees when replacing the reference point to be the (estimated) fixed point and under the convexity assumption.
>
> Note that many existing papers empirically demonstrate the performance of the algorithm when some assumption (e.g., convexity) is violated. Our paper also provides empirical evidence that leveraging the reference point (instead of the fixed point) performs reasonably well (i.e., at least competitive or better than both myopic and uniform baselines) across multiple configurations. We will clarify these points and provide additional discussion about the theoretical connection to existing work in the revision.
>
> [1] Matching of Users and Creators in Two-Sided Markets with Departures.
>
> [2] Performative Prediction in a Stateful World.
>
> [3] How to Learn when Data Gradually Reacts to Your Model.
>
> ---
>
> > **What additional insight do we get from multiple simulation experiments?**
>
> This is to show that the look-ahead policy performs adaptively well in the case where the myopic succeeds and fails. As we can see in the experiment result, both uniform and myopic can be a good choice in one of the scenarios, however, fail catastrophically in the opposite scenario. In contrast, because the proposed approach can care about both myopic outcome (i.e., viewer satisfaction) and investment for provider population growth (i.e., exposure) by design, ours works adaptively well across two opposite situations. Our real-data experiment also provides an example of how an actual population effects can arise in real data. Reviewer 54YD acknowledges our contribution on this point as “it's good that the authors evaluate their method on both synthetic and real-world data.”
>
>
> **(contributions)**
>
> ---
>
> > **that platforms should attend to producer welfare as well as consumer welfare, is reasonably intuitive and the paper fails to explore more implications of their model beyond this basic point**
>
> Thank you for sharing your concerns, and we would like to kindly emphasize that, even if the outcome seems reasonably intuitive, highlighting the overlooked problem is crucial for scientific research.
>
> Regarding this point, we share a similar contribution to existing papers: e.g., [4], which points out the importance of guaranteeing provider exposure under the modeling of viewer/provider departure. We generalize the discussion in the setting where the viewer/provider both increases and decreases their population (i.e., a more complex situation), and demonstrate that caring about producer welfare is indeed beneficial.
>
> Our paper provides the theoretical implications and evidence for the platform designers that caring for provider welfare is indeed crucial for the long-term success of the platform, not only for the ethical perspectives. Reviewer 54YD also acknowledges this point “I believe this work contributes to the broad literature on two-sided platform. It touches on important subjects including balancing the interests of strategic agents and balancing the tradeoff between short-run and long-run welfare.”
>
> [4] Optimizing Long-term Social Welfare in Recommender Systems: A Constrained Matching Approach.

---

### Decision · Program_Chairs · 2025-05-01

**Decision:**

Accept (poster)

**Comment:**

This paper studies population dynamics of two-sided matching platforms. For the most part, reviewers found the model and message of the paper (that myopic greedy policies can lead to sub-optimal welfare while the simple look-ahead policy empirically performs much better) interesting, and the experiments generally convincing. However at the same time, some reviewers felt that the theoretical contributions of the paper were marginal and would have appreciated a deeper theoretical analysis of e.g. the performance of the look-ahead policy, making this a borderline case. We encourage the authors to take the chance and further strengthen their paper regardless of the outcome.